**An Assessment of Antarctic Sea-ice Thickness in CMIP6 Simulations with Comparison to the Satellite-based Observations and Reanalyses**

**Shreya Trivedi[1], Will Hobbs[2], and Marilyn Raphael[1]**

[1]Department of Geography, University of California, Los Angeles

[2]Australian Antarctic Program Partnership, Institute for Marine and Antarctic Studies, University of Tasmania, nipaluna/Hobart, Australia

**Corresponding author:** Shreya Trivedi (shreyatrivedi26@ucla.edu)

**Abstract**

Sea-ice thickness, though critical to our understanding of sea-ice variability, remains relatively understudied compared to surface sea-ice parameters in the Southern Ocean. To remedy this, we examine spatio-temporal variations in sea-ice thickness by analyzing *historical* simulations from 39 coupled climate models in CMIP6, comparing them with three sea-ice products, including satellite-derived observations and reanalyses. Furthermore, we compare seasonal trends in simulated sea-ice thickness with trends in sea-ice area. Our results indicate that CMIP6 models can replicate the mean seasonal cycle and spatial patterns of sea-ice thickness. During its maximum in February, these models align well with satellite-based observation products. However, during the annual minima, CMIP6 models show significant agreement with the reanalysis products. Certain models exhibit unrealistic historical mean states compared to the sea-ice products resulting in significant inter-model spread. CMIP6 models can simulate sea-ice area more accurately than the sea-ice thickness. They also simulate a positive relationship between the two parameters in September such that models with greater area tend to exhibit thicker ice. In contrast, there is a negative relationship in February when greater area is associated with lower thickness since only the thicker ice survives the summer melt. Moreover, our study highlights significant positive trends in sea-ice thickness observed during the cooler seasons, which are nearly absent in the warmer seasons where positive trends are predominantly observed in sea-ice area. The spatial distribution of SIT biases is closely linked to uncertainties in modeling the ice edge and the dynamic processes, emphasizing the need for better model representation of both. This study, therefore, highlights the need for improved representation of Antarctic sea-ice processes in models for accurate projections of thickness and related volume changes.

## 1. Introduction

Antarctic sea-ice extent which showed a small positive linear trend since the start of the satellite era (Cavalieri & Parkinson, 2008; Parkinson & Cavalieri, 2012; Turner et al., 2015; Zwally et al., 2002), has decreased significantly since mid-2015 (Eayrs et al., 2021; Raphael and Handcock, 2022; Wang et al., 2022; Turner et al., 2022). Since there is a long and reliable observational record available for the surface characteristics (such as extent and area) of the ice, they have been the primary focus for understanding the variability in the sea-ice cover in the Antarctic. However, complete understanding of the changes in sea-ice and their potential impact on climate (*via* a variety of climate-sea-ice feedbacks) and marine ecosystems is not possible without an understanding of the variability in sea-ice thickness (SIT) and volume (SIV) (Holland et al., 2006; Stammerjohn et al., 2008).

SIV (*viz.* the product of both area and thickness) serves as a measure of total sea-ice production and, hence, a measure of the surface salinity flux in winter, the freshwater input to the ocean in summer, and total heat exchange with the atmosphere (Maksym et al., 2012). Understanding the variability in SIV improves our understanding of surface buoyancy flux and its impact on the ventilation of Southern Ocean deep waters, as well as trends and variability in salinity in the region (Haumann et al., 2016; Abernathy et al, 2016; Pellichero et al., 2018). This understanding in turn, informs our knowledge about global ocean heat and carbon uptake processes (Williams et al., 2023).

SIT varies seasonally and is an important component of the Antarctic ice budget (Kurtz & Markus, 2012; Worby et al., 2008). SIT is important for the marine biology of the Antarctic ecosystem. It affects the maximum biomass of algae in different ice layers, which in turn influences the food web of the Southern Ocean. SIT, along with the snow depth, affects the light penetration and availability for the phytoplankton contributing further to their production and bloom (Massom & Stammerjohn, 2010; Schultz, 2013). Therefore, a long-term assessment of SIT/SIV in combination with surface sea-ice characteristics, is important for a complete understanding of the ongoing changes in the Antarctic sea-ice and its impacts (Massonnet et al., 2013; Sallée et al., 2023).

Among the currently available SIT datasets, ship-based observations from ASPeCt (Antarctic Sea ice Processes and Climate) tend to underestimate mean measurements because ship routings preferably avoid thicker sea-ice (Worby et al., 2008). Airborne electromagnetic data, NASA Operation *IceBridge* (Koenig et al., 2010) and the upward-looking sonars (ULS) (Behrendt et al., 2013) provide valuable spatio-temporal information but are limited to certain regions, lacking circum-Antarctic distribution. To overcome some of the limitations, advanced retrieval techniques in the form of satellite remote sensing, including passive microwave sensors for thin ice (Kurtz & Markus, 2012), and active sensors like Synthetic Aperture Radar (SAR), have now been applied to study circum-Antarctic SIT coverage and its long-term trends. More recently published satellite-derived (both radar from Envisat-CryoSat-2 and laser from ICESat-2) altimetry-based SIT datasets have proven to be the best source for circum-Antarctic SIT retrievals over the full thickness range (Kurtz and Markus, 2012; Kacimi and Kwok, 2020; Wang et al., 2022). In general, gathering Antarctic SIT data presents significant challenges owing to harsh weather conditions, extensive snow cover, and intricate snow metamorphism processes. These factors introduce uncertainties in both *in-situ* measurements and satellite altimetry observations, particularly concerning the accuracy of detecting the snow-ice interface in the latter method. *In-situ* measurements like drilling data are accurate but extremely limited in time and space and suffer from biases. We discuss such uncertainties in detail in Sect.2.1.

Given the inconsistencies and limitations in the existing SIT observations, Global Coupled Climate Models (GCMs) can serve as potentially valuable tools for assessing long-term SIT/SIV variability and providing future projections. Hou et al. (2024) compared SIT simulations in CMIP6 models with radar altimetric datasets (Envisat-CryoSat-2) and identified common issues, including lags and significant underestimation in SIT climatology during autumn and winter months, as well as negative biases, particularly in the sea-ice deformation zone around Antarctica. In an earlier study, Trivedi et al. (2024) carried out a preliminary comparison of Antarctic SIT across CMIP6 models using *historical* simulations (1979–2014). However, that study only provided a brief analysis of SIT trends and inter-model variability. A detailed assessment of model biases, seasonal behavior, model performances in warmer scenarios and connections with other sea-ice parameters is lacking. An assessment of the accuracy of SIT simulations in GCMs remains a challenge, which adds to the *low confidence* in Antarctic sea-ice projections (Meredith et al, 2019). This is a key motivation for our study.

In this study, we present a high-level evaluation of models in the Sixth Coupled Model Intercomparison Project (CMIP6; Eyring et al., 2016) which simulate Antarctic SIT. We compare these simulations to three different sea-ice products, including radar altimetric (as in Hou et al., 2024) as well as two reanalysis (synthesis) datasets. Our results indicate that CMIP6 models can reasonably capture the timing of the annual cycle and the spatial patterns in SIT, albeit with some

biases and model discrepancies, which we discuss. Notably, when compared to sea-ice area (SIA), the models' performance remains suboptimal. This underscores the need for further improvements in the representation of sea-ice dynamics and the physical processes controlling sea-ice-ocean interactions in GCMs.

Since our understanding of historical sea-ice variability and the evaluation of climate models has traditionally relied heavily on SIA records, this study extends the analysis by comparing simulated SIT and SIV to SIA, in order to see how these diagnostics are related. Furthermore, we examine the inter-relations between SIT and SIA during two key months to better understand their covariability. We also identify significant seasonal trends in the three sea-ice parameters and analyze their evolution over the selected time-period. These results emphasize the importance of incorporating SIT metrics into future model evaluations to enhance our understanding of Antarctic sea-ice dynamics and improve sea-ice and climate projections.

## 2. Data and Methods

### 2.1 Sea-ice products

Our study uses three different sea-ice records for SIT (Table S1): A dataset derived from Envisat-CryoSat-2 (2002-2017; henceforth referred to as satellite product), the Global Ice-Ocean Modeling and Assimilation System (GIOMAS, 1979-2014) and the German contribution to the Estimating the Circulation and Climate of the Ocean project Version 3 (GECCO3, 1979-2014). We compare them with the CMIP6 *historical* simulations which run until 2014. Since the satellite products do not begin before 2002, our study focuses on the time-period between 2002-2014. To eliminate mismatch in the spatial resolution, all the sea-ice products were regridded onto a common CMIP6 model grid. SIA is calculated by multiplying the monthly values of sea-ice concentration (SIC) by the corresponding grid cell area and summing over the Southern Hemisphere. SIV was computed as the product of the actual floe thickness, SIC and the grid cell area, summed over the circum-Antarctic region.

*SIT from Satellite Altimetry- Envisat and CryoSat-2:*

The Sea-Ice Climate Change Initiative (SICCI) project provides a large-scale Antarctic SIT dataset from Envisat and CryoSat-2 with a 50 km spatial resolution (Hendricks et al., 2018a, 2018b). Both Envisat and CryoSat-2 carry a radar altimeter based on the Ku Band frequency which is expected to measure the ice freeboard (total freeboard minus snow depth), with spatiotemporal resolution and spatial coverage consistent with each other. Our study utilizes the aggregated time-series data from both satellites owing to the reasonable alignment of mean and modal values derived from their radar freeboards throughout the sea-ice seasons (Hou et al., 2024; Schwegmann et al., 2016). The SICCI dataset stands out as the most comprehensive satellite dataset spanning the circum-Antarctic SIT from 2002 to present (Shi et al., 2021, Hou et al., 2024). This coverage is not matched by more recent datasets like ICESat-2 (Xu et al., 2021). It is comparable to ULS-derived SIT for the Weddell region (Shi et al., 2021; Liao et al., 2022; Wang et al., 2022) and aligns well with *in-situ* ship-based observations (ASPeCt; Worby et al., 2008), showing highest thickness in summers and lowest in autumn-winter.

Retrieval of such Antarctic SIT products is quite challenging and the existing satellite datasets are
not without uncertainties. Although the altimetric SIT measurements of Envisat and CryoSat-2
observations represent a major advancement in monitoring Antarctic SIT, it is an experimental
climate data record with uncertainties resulting from the inaccuracy in determining the snow–ice
interface (Willatt et al., 2010) and biases due to surface-type mixing and surface roughness
(Schwegmann et al., 2016; Paul et al., 2018; Tilling et al., 2019) resulting in overestimations
(Hendricks et al., 2018a, b; Shi et al., 2021; Wang et al., 2022).

For the satellite product, SIA is calculated by multiplying the SIC with the corresponding grid cell
area, both derived from the reanalysis dataset provided by the National Snow and Ice Data Center
(NSIDC; Comiso, 2017) for the period 2002-2014. SIV is then computed by multiplying the
resulting SIA with the satellite-derived floe thickness and summing over the entire Southern
Hemisphere.

*SIT from Reanalysis/Synthesis products:*

Reanalyses integrate information from observations and ocean-sea-ice models through data
assimilation and provide gridded sea-ice data with homogeneous spatiotemporal sampling over an
extended time-period. The use of ocean–sea-ice models forced by atmospheric reanalysis is a
general approach to better constrain SIT changes with the observations, in both the Arctic and
Antarctic. Since the reanalysis datasets offer state estimations closer to observations compared to
model-only data, it makes them a valuable tool in Antarctic sea-ice studies (Kumar et al., 2017).
Hence, our study uses two such estimates of long-term Antarctic SIT changes and compares them
against the satellite products and the GCMs.

GECCO3 ocean synthesis is an improved version of GECCO2 based on MITgcm which employs
the adjoint method to fit the model to a large variety of data over a multidecadal period (1948-
2018). Unlike GECCO2, it is optimized over only one assimilation window. GECCO3 has 40
levels and uses the horizontal and vertical grid of the ocean component of MPI-ESM in the MR/HR
configuration, providing a global eddy-permitting synthesis at a nominal resolution of 0.4° (Köhl,
2020). While GECCO3 does not directly assimilate sea-ice data, its ocean-ice coupling is
influenced by the model's oceanic state. Hence, this synthesis primarily assimilates oceanic data,
including temperature and salinity (EN4.2.1), along-track sea-level anomalies (AVISO), and sea
surface temperature (HadISST), using a 4D-VAR (adjoint) method (Köhl, 2020).

GIOMAS uses the Parallel Ocean Model coupled with a 12-category thickness and enthalpy
distribution ice model at a horizontal resolution of 0.8° (Zhang & Rothrock, 2003). In GIOMAS,
the modeled SIC is nudged by assimilating satellite-derived SIC from the Special Sensor
Microwave Imager (SSM/I) launched by the Defense Meteorological Satellite Program (Weaver
et al., 1987). Subsequently, other modeled variables including SIT are adjusted accordingly. This
process reduces the root-mean-square difference and improves the correlation between modeled
SIT and observed SIT, while also causing the thinning of the mean SIT. This assimilation method
has demonstrated good agreement of modeled SIT with satellite observations in the Arctic
(Lindsay & Zhang 2006) and is useful for studying long-term variations in Antarctic sea-ice (Liao
et al., 2022; Shi et al., 2021).

To make the GECCO3 and GIOMAS products comparable to absolute floe thickness estimates (the SIV per grid-cell area or "equivalent sea-ice thickness"), we convert them into "effective thicknesses" by dividing them by their respective SIC records. For the reanalysis/synthesis datasets, SIA is calculated as the product of the SIC and the corresponding grid cell area obtained from the respective datasets. The SIA value so obtained is then multiplied with the "effective thickness" from each of the reanalysis/synthesis dataset to obtain their respective SIV.

## 2.2 CMIP6 Models

We analyze the *historical* experiments of the CMIP6 dataset, specifically focusing on the *sithick* variable, which represents simulated effective or actual floe thickness. We also use *siconc* (sea-ice concentration) and *areacello* (area of individual grid cells over the ocean) variables. CMIP6 models generate multiple ensemble members, which are multiple runs or simulations with slightly different initial conditions or parameter settings, used to capture uncertainty and variability in model predictions. In this study, we consider a single ensemble for each model (Table S2) to account for internal variability and ensure fairness by not giving weight to the models with multiple ensemble members (following Notz & Community, 2020; Roach et al., 2020). We calculate SIV by multiplying *siconc*, *sithick* and *areacello* and summing over the circum-Antarctic Southern Ocean. For SIA, we calculate the area integral product of *siconc* and *areacello*. Lastly, for floe thickness, we use the averaged *sithick* over the Southern Ocean. The multi-model means (MMM) are calculated based on the single ensemble member in 39 coupled climate models for all the three sea-ice variables. In addition to the *historical* experiments, we also analyze a warmer scenario, *SSP 5-8.5* (Shared Socio-economic Pathway; 2015-2100) to compare with the future sea-ice variability.

## 3. Results

### 3.1 Mean and Anomaly State

This section first discusses the mean annual cycles of SIA, SIV and SIT for the different sea-ice products and the CMIP6 models over the period of 2002-2014 (Fig.1). It then examines the biases and spreads in the climate models and, finally, the simulated and observed seasonal trends in the anomalies across all the sea-ice variables.

Fig.1a highlights that all the sea-ice products agree on the timing of their SIA maxima and minima (in September and February, respectively). However, GIOMAS, despite having the same maximum and minimum timings, exhibits a lower amplitude, particularly during the winter season. All the CMIP6 models similarly simulate the observed timing of maxima and minima in SIA, however the MMM remains consistently lower than all other sea-ice products throughout the year. Overall, significant negative biases are observed in the simulated SIA cycles, except for a few models that consistently simulate larger SIAs throughout the year.

For the SIV, all the sea-ice products display pronounced annual cycles (Fig.1b). While their minima are similar in timing and magnitude, their maxima do not occur at the same time, with the satellite products showing the earliest maxima and GECCO3 the latest. Additionally, the amplitudes of the SIV cycles are highest in the satellite products and lowest in GECCO3. All the CMIP6 models simulate a similar annual cycle to the sea-ice products, with their MMM maxima in October and minima in March (lagging the sea-ice products by one month). Like SIA, the MMM SIV is biased low compared to all the sea-ice products, agreeing best with GECCO3.

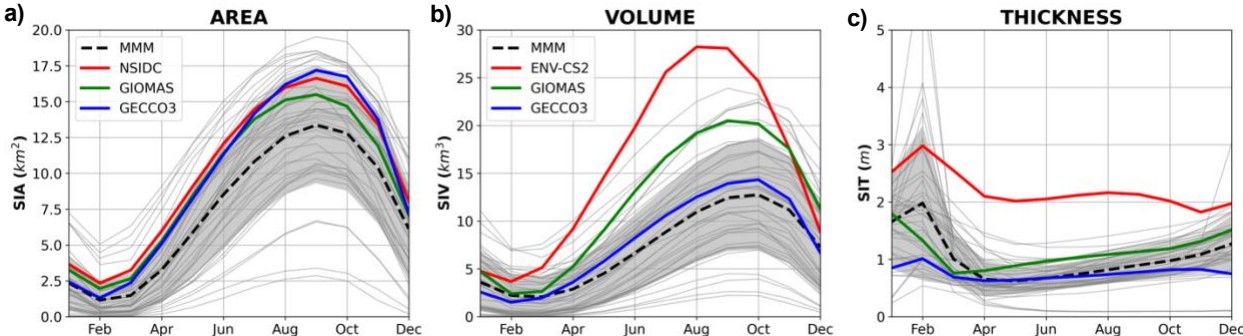

**Figure 1: Comparison of annual cycles of SIV, SIT and SIA of the circumpolar Antarctic. All the CMIP6 models are shown as grey lines, The Multi-Model Mean (MMM) is the black dashed line. GECCO3 in blue, GIOMASS in green, and Envisat-CryoSat-2/NSIDC in red. Grey shaded areas are +-1 standard deviation for the MMM. Scale: million km² and thousand km³.**

Fig.1c shows that SIT has a different annual cycle than SIA and SIV. Its maximum occurs in February, which is observed across all the sea-ice products, except for GIOMAS, where the sea-ice maximum occurs one month earlier. The timing of the SIT minimum is more variable. For the synthesis products, the SIT minimum occurs at the beginning of the growth season, whereas for the satellite products it occurs during the retreating season. CMIP6 models can capture the thickness maxima in February and agree with the synthesis products in showing minima in the fall. However, there is considerable spread in the exact timing of the minimum, with the MMM minimum occurring in May, lagging the reanalysis products. Most of the models simulate thicker sea-ice than the three sea-ice products during summer (between January-April) resulting in biases which considerably decrease for the synthesis products and increase for satellite datasets starting in May. A few models, namely the IPSL and EC-Earth3-models (characterized by significant warm Southern Ocean biases; Döscher et al., 2022), exhibit anomalously thick sea-ice (>3m) in February. This can be attributed to the dynamically-related reasons causing the drift and deformation of the ice. This thick ice is captured in these models mainly due to the well-resolved wind patterns (Vancoppenolle et al., 2009) which result in the sea-ice drift and accumulation of the multi-year ice which survives the summer melt and is found mainly in the Weddell Sector during February. Additionally, in models with NEMO-LIM3 sea-ice modules, such as IPSL- and the EC-Earth3-models, the frazil ice formation dominates the total ice growth which may again lead to an accumulated SIT during February (Lie et al., 2021). By contrast, CNRM-models exhibit anomalously low thickness throughout the year. This is primarily due to high negative biases in their SIC simulations and their inability to realistically simulate thick sea-ice during the austral summer in the Weddell Sea (Voldoire et al., 2019). Moreover, these differences can be attributed to various factors, including the multiyear sea-ice surviving the summer melt (Worby et al., 2008), challenges associated with satellite retrieval of summer SIT (Kurtz and Markus, 2012), and the influence of dynamical wind patterns in CMIP6 models, resulting in excessive ridging (Lie et al., 2021).

Summer maxima in SIT vary between January and February depending on the sea-ice product used (Fig.1c). Such high SIT values during summer stand in contrast with the annual minima observed in other sea-ice variables. This SIT annual cycle results from the melt of first year ice in large areas in the seasonal ice zone so that only the thickest ice survives. In the beginning of the freezing

season, large areas are covered by newly formed first year ice, which reduces the mean freeboard
compared to summer values (Schwegmann et al., 2016). Therefore, the highest average thickness
in February is due to the compacted ice which survives the melt season (Kurtz & Markus, 2012;
Worby et al., 2008; Xu et al., 2021). It is for this reason that the SIT seasonal cycles look very
different from those of SIA or SIV. Therefore, to capture the sea-ice seasons based exclusively on
the annual cycle of SIT, we conducted the analyses in Sect. 3.3 and 3.4 using February and
September.

The inter-model agreement also varies considerably between seasons and across variables. The
inter-model spread of annual mean Antarctic SIT, SIV and SIA is 5.9 m, 20 thousand km$^3$ and 16
million km$^2$ for the maxima and 1.8 m, 7.5 thousand km$^3$ and 4 million km$^2$ for the minima,
respectively. Fluctuations in the inter-model spread are larger during fall and winter for SIV and
SIA but shrinks during summers. By contrast, SIT has greater inter-model spread during summer
and shrinks significantly from April-November. Notably, for most parts of the year, the inter-
model spread in SIT remains smaller than disagreements within the sea-ice products, owing to the
overestimations in satellite product during the winter months.

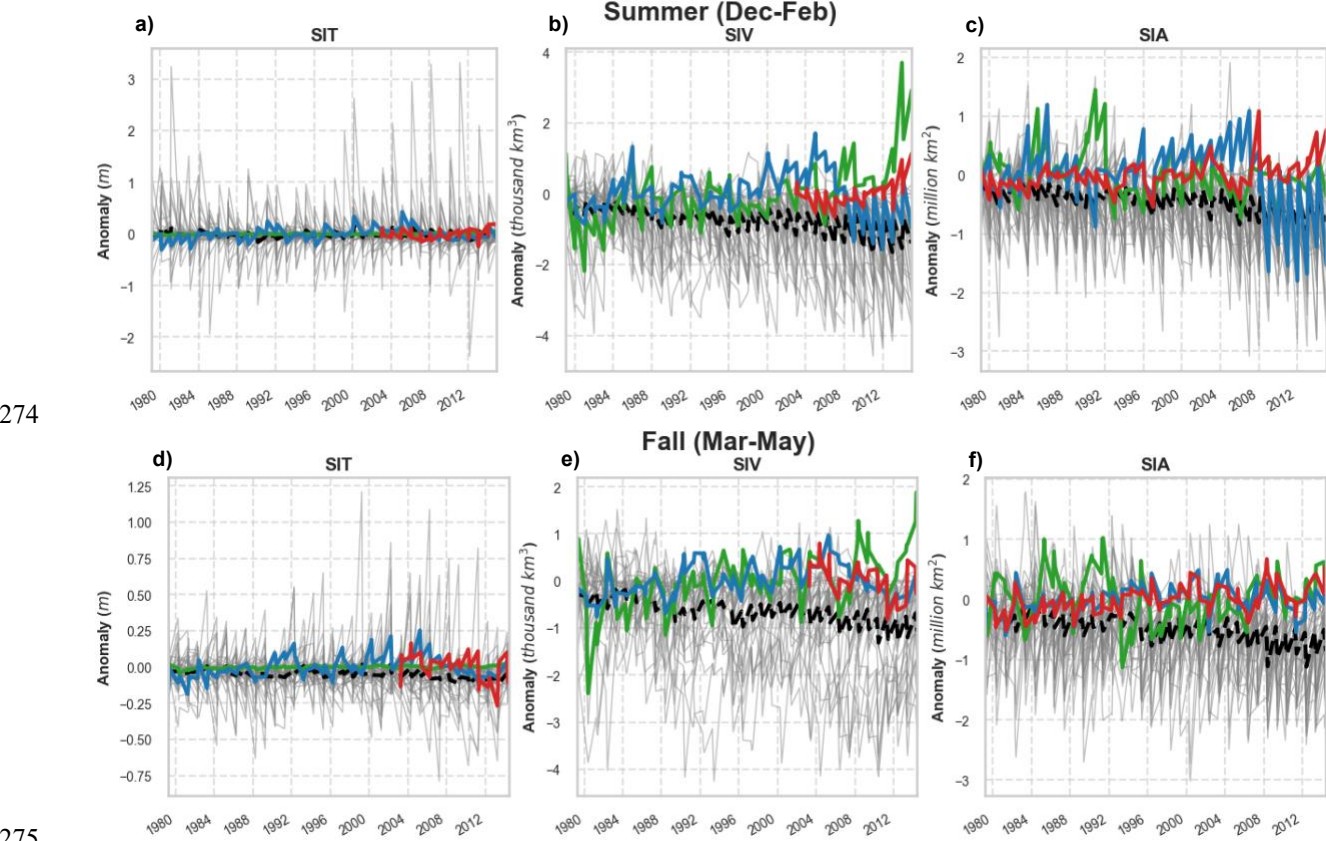

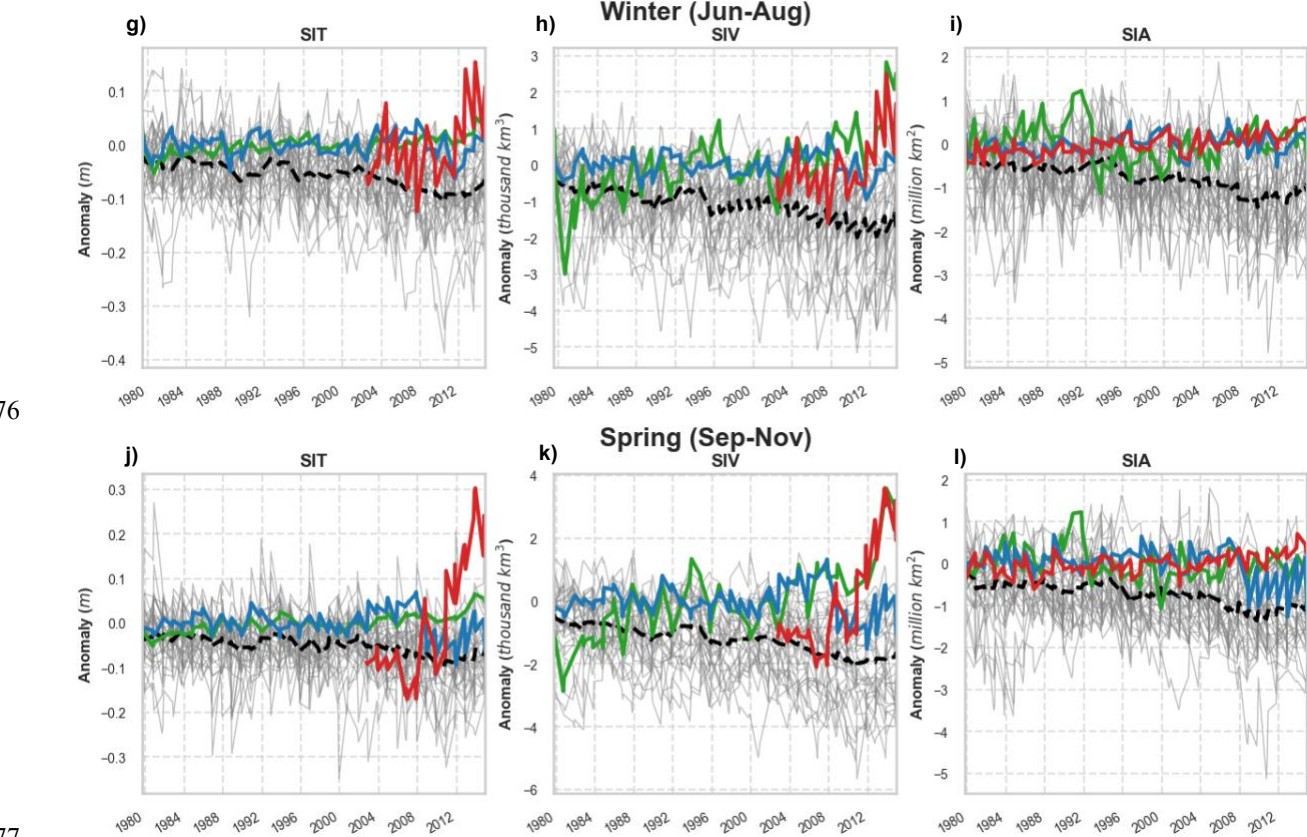

**Figure 2: Anomalies for four seasons: Summer and Fall (Warm Seasons; a-f), Winter and Spring (Cold Seasons; g-l) of SIT (left), SIV (middle) and SIA (right) of the circumpolar Antarctic. All the CMIP6 models are shown as grey lines, Multi-model mean in dashed line, GECCO3 in blue, GIOMASS in green, and NSIDC/ Envisat-CS-2 in red. All the sea-ice products extend for the time-period 1979-2014 except for satellite products which are only available for 2002-2014.**

Next, we look at the seasonal trends in the anomalies of the three sea-ice variables (SIT, SIV and SIA) across different CMIP6 models and the sea-ice products. In general, CMIP6 models simulate negative trends in Antarctic SIA and extent (Roach et al., 2020 and Shu et al., 2020), which contrast with the observed positive trend until mid-2015 (Li et al., 2023; Shu et al., 2015; Turner et al., 2013). Figure 2 demonstrates a similar pattern in the simulated SIT/SIV with a negative trend noticed across all the sea-ice variables. However, differences emerge in seasonal trends for different sea-ice products (blue lines for GECCO3; green for GIOMAS and red for the satellite products in Fig.2) starting in the early 2000s.

In particular, significant positive trends in SIT/SIV are observed during the cooler seasons (winter and spring; Fig.2g-h,j-k), while no such trends (or very weak trends in SIV) are detected in the warmer seasons (Fig.2a-b,d-e). For SIA, small positive trends are observed during the warmer seasons (Summer and Fall), consistent with the previous studies (Martinson, 2012; Eayrs et al., 2019), while these trends are markedly reduced during the cooler seasons. This reduction is attributed to the thermodynamic constraints imposed by the Southern Ocean polar front, which limits ice edge expansion during the colder months. The observed positive trends in SIT/SIV during winter and spring imply that changes in Antarctic SIT may contribute to variability in total

sea-ice mass/volume during these colder months. This could be linked to the presence of robust
land-ocean temperature gradients during winters, which may result in high-intensity winds, one of
the significant contributors to SIT/SIV fluctuations in the Southern Ocean (Zhang, 2014).

## 3.2 Future Projections under *SSP5-8.5* (2015–2100)

Since our study uses *historical* experiments in climate models which ends in 2014 (due to the cut-
off limits of the *historical* simulations), this section acknowledges the importance of assessing
future sea-ice developments, particularly in the Antarctic where pronounced declines in the sea-
ice have been observed since 2016 (Eayrs et al., 2021; Raphael and Handcock, 2022; Wang et al.,
2022; Turner et al., 2022).

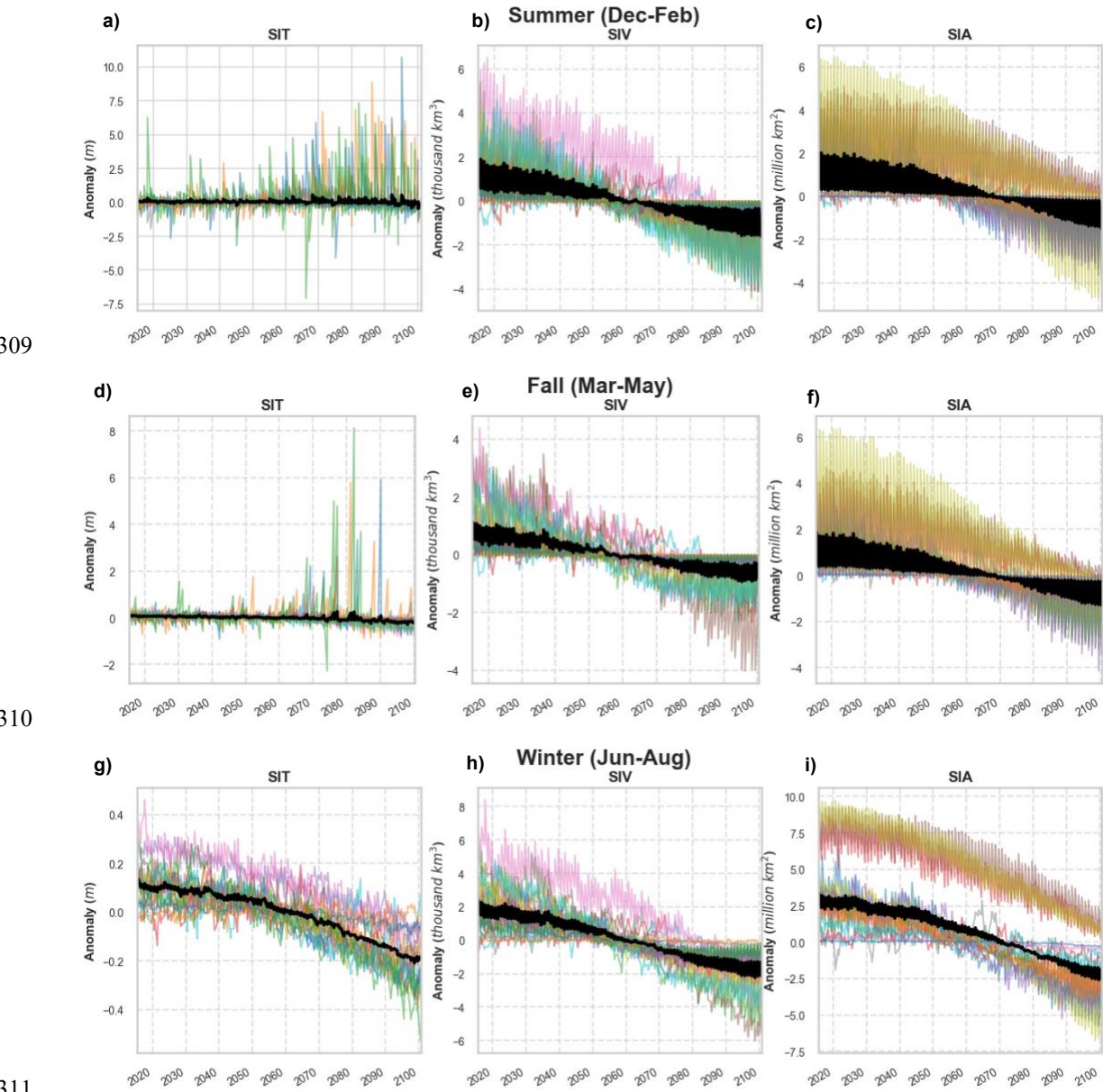

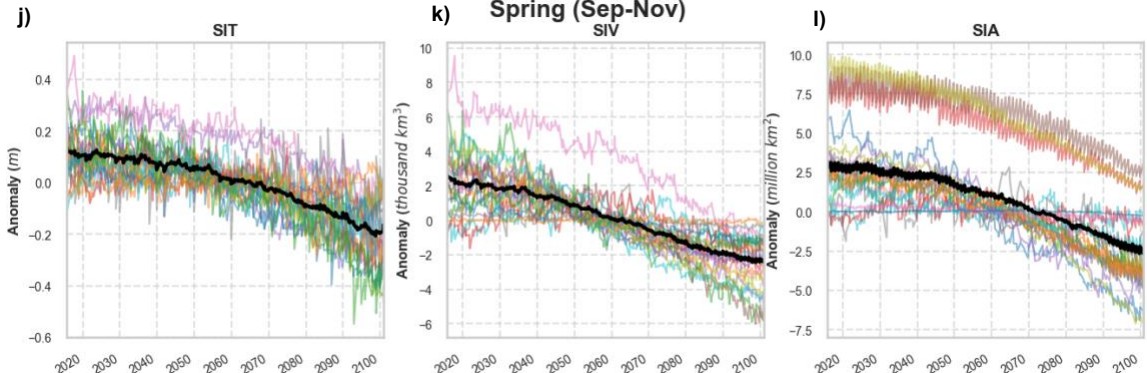

**Figure 3: Anomalies for four seasons: Summer and Fall (a-f; Warm Seasons), Winter and Spring (g-l; Cold Seasons) of SIT (left), SIV (middle) and SIA (right) of the circum-Antarctic. All the CMIP6 models are shown as colored lines, and the Multi-model Mean in black line. The time-period is 2015-2100 for the *SSP5-8.5* scenario. Since sea-ice products end in 2014, they could not be included here. Note that 3 models (namely ACCESS-CM-2, ACCESS-EM2-1 and CESM2-WACCM show slightly higher anomalies (especially for SIA), particularly evident during the cooler seasons (i and l).**

We examined the seasonal anomalies in sea-ice variables under the high-emission *SSP5-8.5* scenario for the period 2015–2100 (Fig. 3). The results are consistent with those in Fig. 2, showing pronounced seasonal trends in SIT, particularly during the cooler seasons (winter and spring). Under the warmer scenario, all CMIP6 models project a notable decline in SIT during these cooler months, while trends in SIT are largely absent during the warmer seasons (summer and fall). In contrast, SIV and SIA exhibit a persistent year-round decline, with negative anomalies becoming more pronounced from approximately 2060 onward (Fig. 3).

When assessing the Antarctic sea-ice distributions in the post-2014 period, it is expected that under warming scenarios, models will show reductions in sea-ice owing to their response to increasing temperatures. However, our results reveal a seasonally asymmetric pattern of decline: SIA and SIV decrease persistently throughout the year, while SIT exhibits notable thinning only during cooler seasons. This indicates that the overall reduction in Antarctic sea-ice projected under warmer scenarios is likely driven by sustained losses in area and volume as actual thinning is not consistently observed across all seasons. Consequently, much of the reduction in simulated sea-ice likely arises from changes in surface coverage rather than from widespread structural thinning. These projected declines also correspond well with the observed post-2016 record-low Antarctic sea-ice extents (Turner et al., 2017; Schlosser et al., 2018), indicating that the recent losses may represent the early onset of the long-term downward trend simulated under high-emission scenarios. Such consistency between observations and projections highlights the increasing vulnerability of the Antarctic sea-ice system to ongoing atmospheric and oceanic warming. Future studies are needed to investigate this in detail.

### 3.3 Seasonal Variations and Inter-relationships

An accurate spatio-temporal distribution of SIT is key to accurate estimates of SIA and eventually SIV distributions. It reflects the skill in simulation of local processes, coupled interactions and

energy transfer among the ocean below, the sea-ice, and the atmosphere above (Stroeve et al., 2014). To estimate seasonal variations and evaluate the performance of the CMIP6 models in capturing the observed distribution of sea-ice variables, we plot the Taylor Diagrams (Fig.4) representing the spatial correlation coefficients, Root Mean Square Deviation (RMSD; not shown in the Figure but included in Tables S3-S5) and standard deviation among 39 models, and the three sea-ice products. We use the satellite dataset as the observation reference for SIT and SIV, calculating RMSDs and correlation values across the grids. For SIA, the reference dataset used is NSIDC. All the calculations were performed based on area-integrated spatial averages of reference datasets of SIT, SIV and SIA over the circum-Antarctic for February and September.

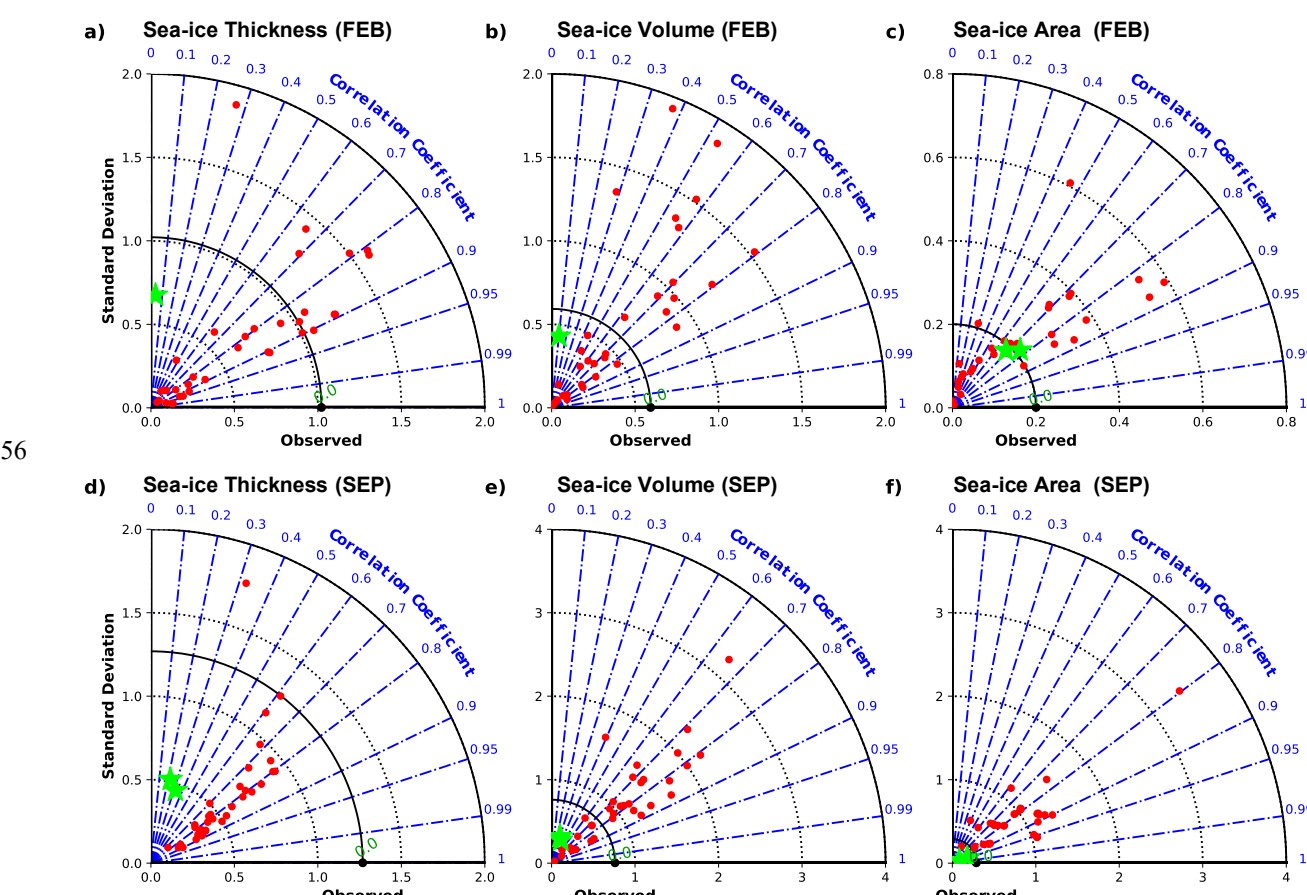

**Figure 4: Taylor Diagrams representing spatial correlation and standard deviation using time-averaged means between CMIP6 models and different sea-ice products. For each model and synthesis product, two statistics are plotted: the Pearson correlation coefficient, related to the azimuthal angle (blue contours); and the standard deviation viz proportional to the radial distance from the origin (black dotted contours). Black solid contour corresponds to the standard deviation of the reference dataset. Red dots represent individual CMIP6 models and green stars represent synthesis datasets. The period used for comparison is 2002-2014 for February (a,b,c) and September (d,e,f). For February (a and b), GECCO3 is not included as it had very small negative correlation coefficients. Reference datasets used for SIT/SIV and SIA are Envisat-CryoSat-2 and NSIDC,**

**respectively. Higher correlation coupled with a lower RMSD indicates greater accuracy of CMIP6 models in simulating the sea-ice variables.**

The correlation coefficients in Fig.4 range between 0.6-0.9 for all the variables in both the months (Tables S3, S4 and S5). Specifically, models tend to exhibit the highest variability in SIT, as indicated by higher RMSDs and standard deviation values that deviate significantly from those of the reference dataset (Fig.4a,d). Both synthesis products (GIOMAS and GECCO3) show the highest variability and lowest correlations for SIT, indicating lack of agreement among different sea-ice products. Whereas the opposite is observed for SIA where most of the models and reanalysis products have a lower variability for SIA (Fig.4c,f) compared to SIV and SIT. When comparing the two months, lower variability and higher correlations are observed for all the sea-ice variables in February (Fig.4a-c). In summary, the Taylor Diagram reveals that while most CMIP6 models demonstrate higher accuracy in simulating SIA and SIV (with high correlations and low RMSDs), their SIT simulations are less accurate. The widely spread standard deviations suggest considerable differences in how the models simulate internal variability in SIT, highlighting their underlying uncertainties in representing such variability. There also exists a disparity in their seasonal accuracy with models performing better during February across all the variables.

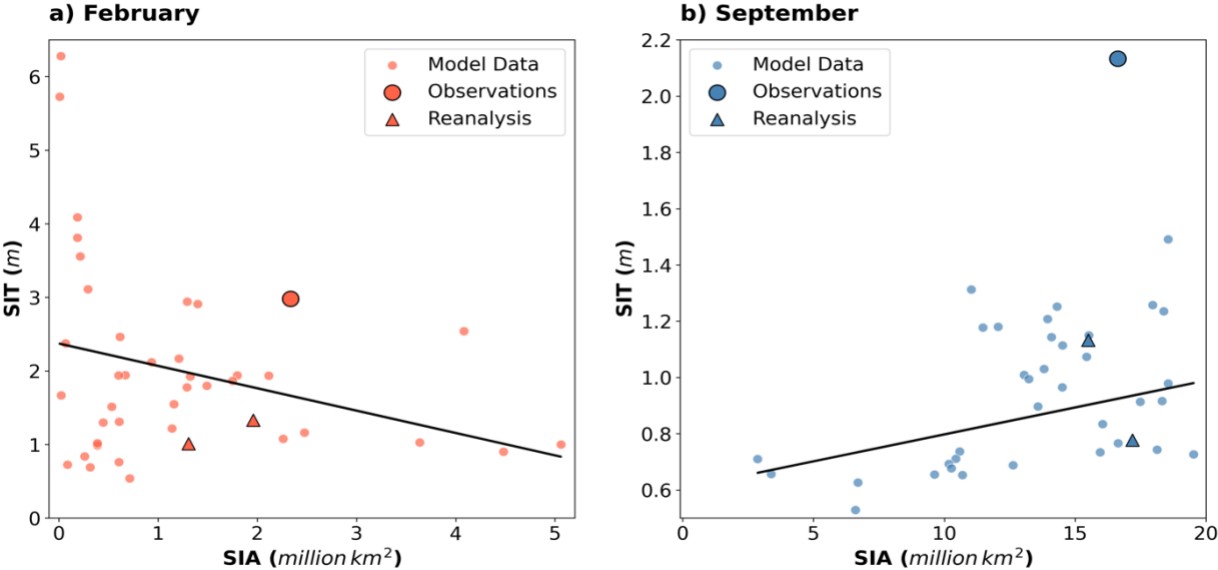

**Figure 5: Scatter plots between the climatological means of SIT (y-axis) and SIA (x-axis) for CMIP6 models and Observations for the period (2002-2014) for February (red) and September (blue). The line of best fit represents the relationship between the two variables for selected months. Each small dot represents a model while the larger dots represent observations (Envisat-CryoSat-2 and NSIDC for SIT and SIA, respectively). The reanalysis/synthesis datasets are represented by triangles. The figure clearly demonstrates seasonal variations in magnitudes of both the variables.**

Given their better performance in simulating SIA, we further investigate whether or not this performance is correlated with SIT accuracy in individual CMIP6 models and the synthesis products. This evaluation has not been done before; however, examining this interrelationship will significantly enhance our understanding of sea-ice changes, which has so far been largely based on surface parameters such as SIA. Additionally, it will provide deeper insights into the

interpretation of existing historical records of surface sea-ice parameters in the Antarctic. For this, we compare the annual averages of SIT and SIA in models and the synthesis products using the satellite product as the observational reference (Fig.5). The comparison shows that SIA biases range between -2 to 2 million $km^2$ during February and between -14 to 4 million $km^2$ during September. In February, although SIT is at maximum, most models simulate thinner sea-ice with values below the satellite estimates of 3m. During September, the models consistently show thin biases with values ranging between 1-2m across all the models.

Figure 5 also highlights how simulated SIA and SIT are differently related depending on the time of year (also evident in Fig.2-3). In summer, SIA and SIT biases are negatively correlated (Fig.5a; although we note that many models have very low SIA in February), and in winter they are positively correlated (Fig.5b). One possible way to explain this is that models with strong melt seasons result in low summer sea-ice cover in which only the thickest ice can survive. The reverse may be true for weaker melt seasons which may allow more, and thinner, ice to survive, leading to greater SIA made up of thinner ice. In winter, greater sea-ice freezing will lead to both thicker ice and a larger area, explaining the positive relationship between the SIA and SIT biases in Fig.5b. This contrasting seasonal relationship raises questions about whether SIA is a reliable predictor of SIT, which has significant implications for our understanding of sea-ice changes when based solely on SIA. Further studies are needed to clarify this.

### 3.4 Seasonality in SIT Biases

This section investigates the seasonality in SIT biases in both temporal and spatial dimensions across three sea-ice products, relative to the multi-model means (MMM) of the selected 39 CMIP6 models. Figure 6 highlights the seasonality in SIT biases, showing patterns in biases with reference to the climatological SIT means across different months and sea-ice products. With respect to the satellite product, most models exhibit negative biases across all months, with the largest bias observed in March. For the two reanalysis/synthesis datasets, the simulated SIT MMM values, except for the February maxima, remain closer to the zero line (within the range of ±0.3m), indicating much smaller biases.

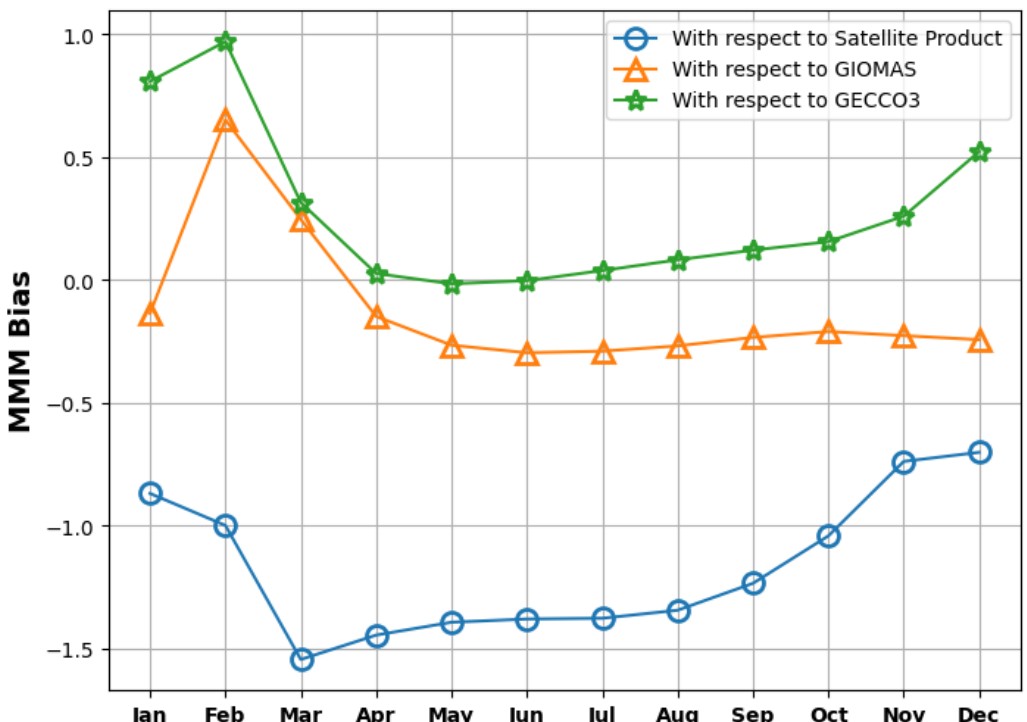

425

**Figure 6: Seasonality in SIT biases (m) calculated as the difference between climatological means of MMM of 39 CMIP6 models and the three sea-ice products (2002-2014).**

This section further explores the spatial spread (measured by the standard deviation of the ensemble) in SIT and SIC biases, using three different sea-ice products for February (Fig.7) and September (Fig.8) over the period 2002-2014. In February, higher standard deviations are evident in the Amundsen-Bellingshausen Seas (ABS) and along the coastal edges of the Weddell Sea (blue shaded regions in Fig.7a), indicating greater model disagreement when compared to the sea-ice products during this month. These regional disagreements may result from dynamic processes such as sea-ice drift, melting, and freezing which are not accurately captured by the models. However, models generally show better alignment with reanalysis/synthesis products in February, as seen in the lower standard deviation values (shown in red), indicating improved agreement.

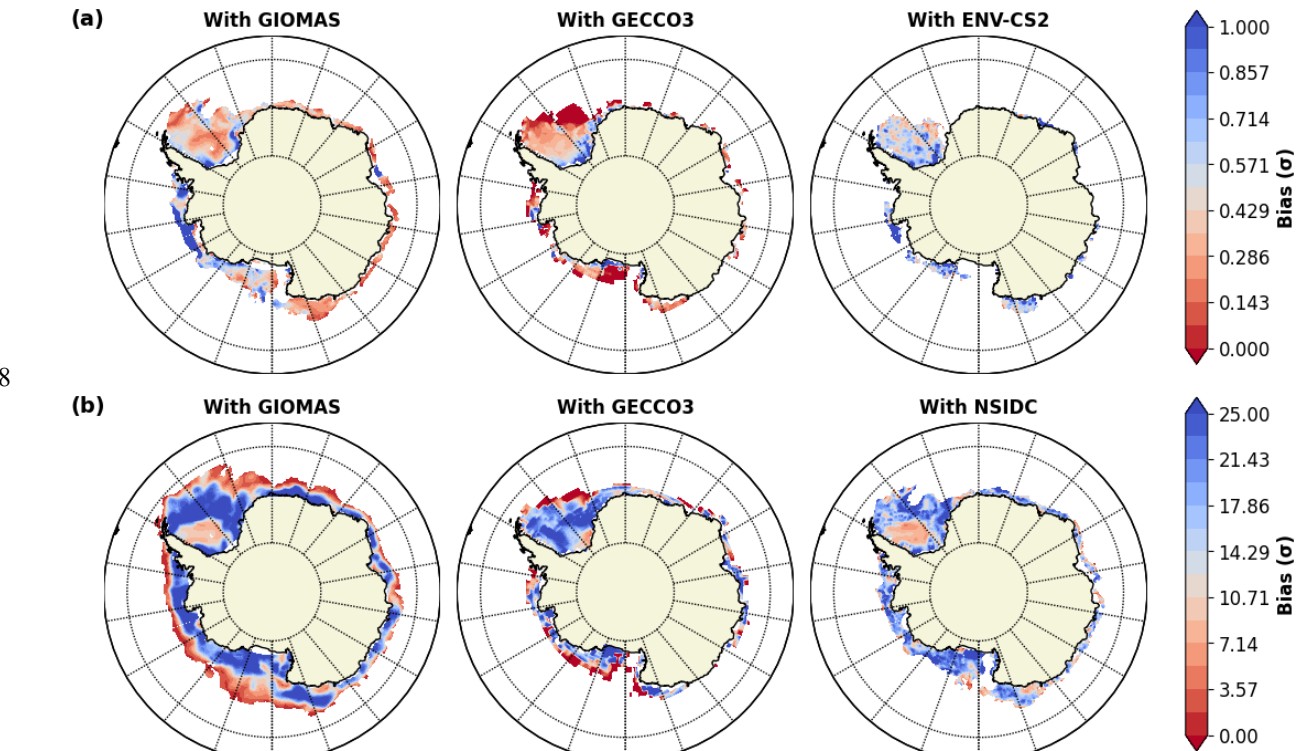

**438**

**Figure 7: Variability of the Spatial Biases in CMIP6 Models (calculated as Standard Deviation of the difference between CMIP6 MMM and the three distinct sea-ice products); for the month of February in (a) SIT (m) and (b) SIC (%).**

In contrast to SIT, the spatial bias spread in SIC is larger (Fig.7b), but the pattern remains similar—better model agreement (lower standard deviations) at the ice edges and in the inner Weddell Sea, while there exists poorer agreement (higher standard deviations) closer to the coasts. Figure 7b also helps to determine how SIC biases may contribute to standard deviations observed in the SIT biases. For instance, the regions of higher model disagreements in SIT that coincide with the high SIC standard deviations suggest that model errors in locating the ice edge may also influence the spread of actual thickness and vice versa. Therefore, when models misplace the ice edge, they might overestimate or underestimate SIT in those regions. In February, the model disagreements in the SIT biases near the ABS and the coastal edges of the Weddell (blue regions in Fig.7a) coincide with higher standard deviation values in SIC bias spread (Fig.7b) in the same regions, especially along the southern Weddell coast. This suggests that some of the SIT biases in these regions could be a result of misrepresentation of the ice edge in models, rather than true variations in actual thickness.

In September, there is an overall lower bias spread in SIT across most of the Antarctic region, except for localized areas along the sea-ice edge where model disagreement is more pronounced, particularly in the ABS and the Eastern Antarctic, and the coastal edges of the Weddell Sea along the Antarctic Peninsula (Fig.8a). When comparing across all the sea-ice products, a relatively higher standard deviation value is noticed in most of the Antarctic regions with respect to the satellite product (Fig.8a). This can be attributed to the satellite product's exaggerated SIT values for September which most models are unable to simulate, leading to greater model disagreements.

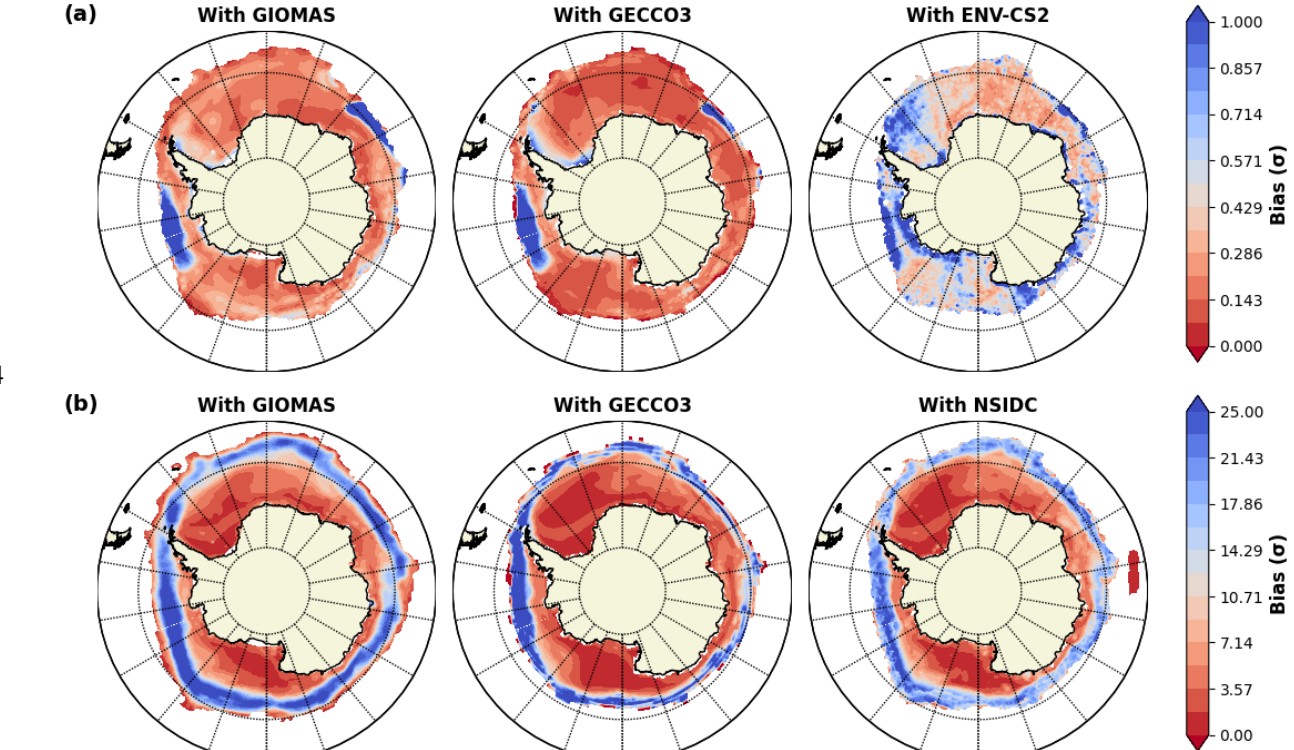

**Figure 8: Variability of the Spatial Biases in CMIP6 Models (calculated as Standard Deviation of the difference between CMIP6 MMM and the three distinct sea-ice products); for the month of September in (a) SIT (m) and (b) SIC (%).**

Figure 8b shows SIC bias spreads for September, where much of the Antarctic region displays low standard deviation values. However, the sea-ice edge remains an area of higher model disagreement, similar to the SIT patterns, suggesting that while the ice interior is well-represented, models continue to struggle with accurately locating the ice edge. The correlation between SIC and SIT bias spreads is again observed in September, where model disagreements along localized ice edges occur in both variables, indicating that SIT biases in CMIP6 models may stem, in part, from uncertainties in where the ice edge is located. However, along the coastal edges of the Antarctic Peninsula, the lower SIC standard deviation values compared to that of SIT suggest that the model disagreements in SIT are more likely driven by the differences in how models simulate dynamic processes (e.g., the Weddell gyre and its role in sea-ice dynamics) rather than by inaccuracies in the ice edge representation. The following section addresses this in more detail across individual models.

### 3.5 Spatial Distributions of SIT

In this section, we perform a spatial comparison for circum-Antarctic SIT across 39 individual CMIP6 model ensembles using different sea-ice products as observational references. However, we only discuss in detail GIOMAS as the observational reference. Here, GIOMAS is selected for a detailed discussion instead of other sea-ice products, primarily due to its relative overall closer agreement with the models (See Sec.3.1) and the very high SIT biases observed in the satellite

products during September (Fig.6 and S1). In general, during both the months the mean spatial distribution of SIT in models shows lower biases with respect to the reanalyses/syntheses when compared to the satellite product (See Sect.3.4). GECCO3 produced results comparable to the GIOMAS reanalysis product (not shown). Although GIOMAS displays an earlier seasonal maximum, the MMMs exhibit the lowest biases relative to GIOMAS for this month, compared to GECCO3 and satellite-derived products (Fig.6).

Spatially, the satellite as well as the reanalysis/synthesis products show that thickest sea-ice resides in the western Weddell Sea along the Antarctic Peninsula and along the coastal edges of the ABS- in the form of multi-year ice (Fig.9 and S1). There is relatively thinner sea-ice observed in the eastern Antarctic (Kurtz & Markus, 2012). Our analysis reveals that most of the CMIP6 models capture a similar spatial pattern in SIT around the Antarctic however, they do exhibit biases and underestimate SIT (Fig.9 and Fig.S1-S3). In February, over half of the models simulate thinner sea-ice in the Weddell Sea compared to the GIOMAS product (Fig.S2) while thin biases relative to the satellite products are simulated almost across all the models (Fig.S3). On the circum-Antarctica scale, about 38% of the models simulate thicker ice compared to GIOMAS, with only 5 out of 39 models showing SIT greater than that estimated by satellite products.

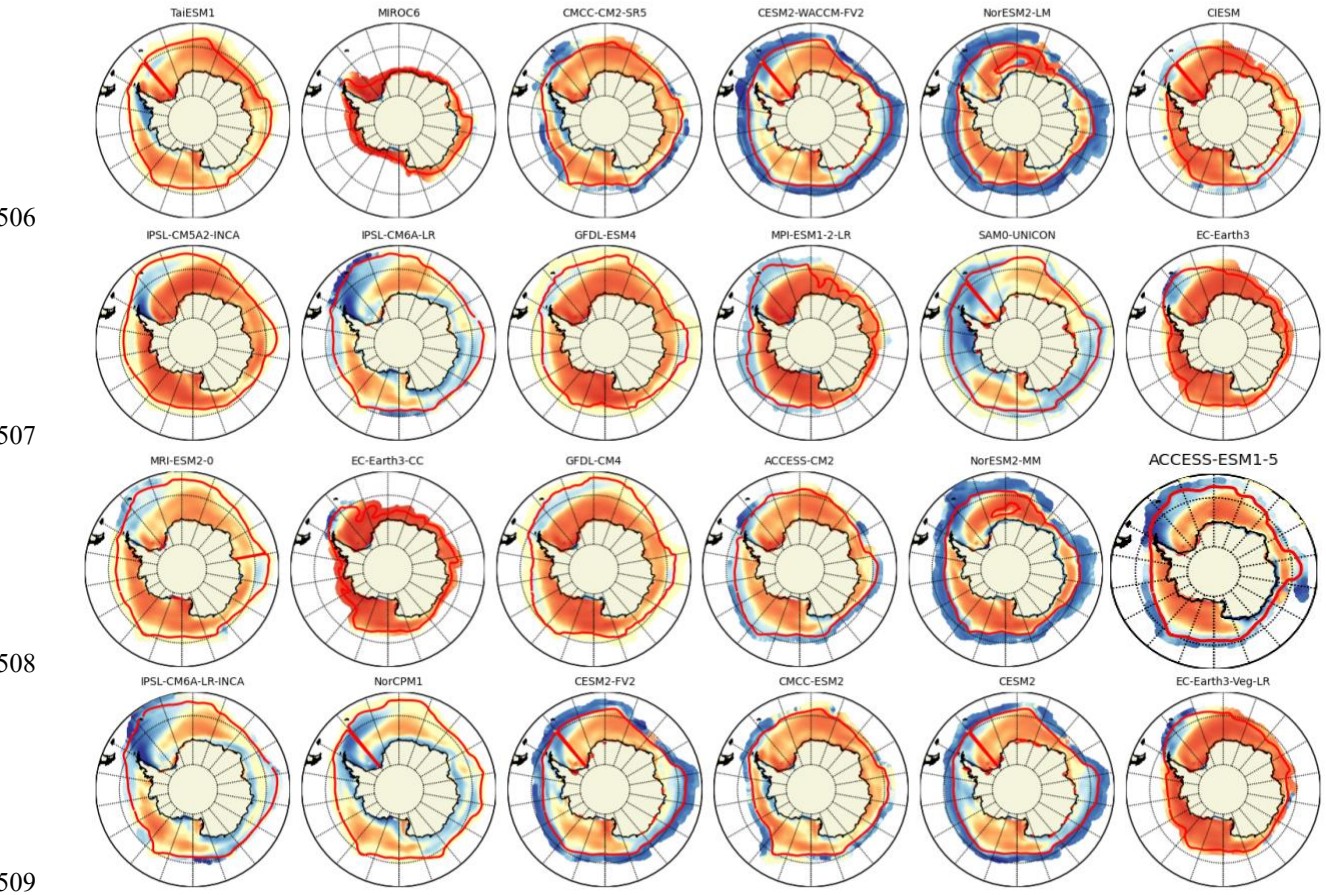

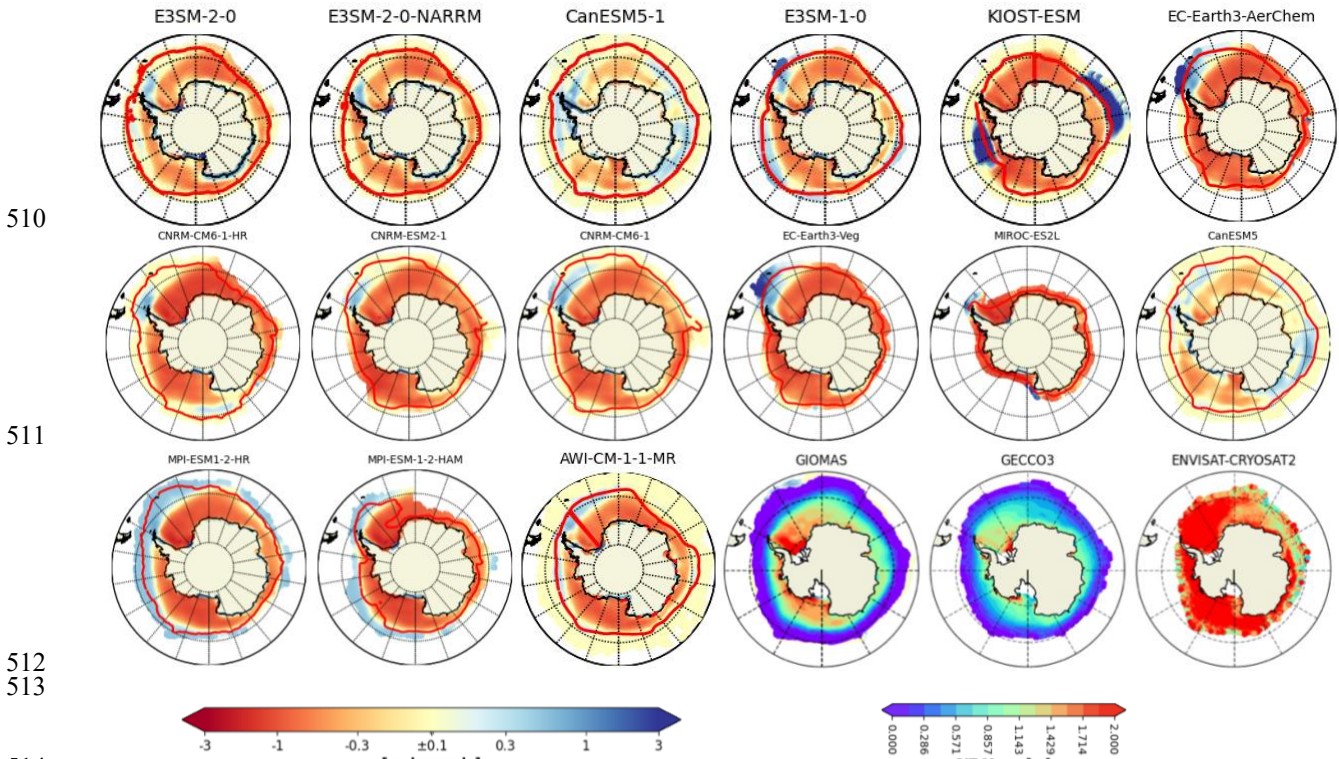

**Figure 9: Spatial Biases of SIT averaged over 2002 to 2014 (September) for 39 CMIP6 models from the reference dataset: GIOMAS. The last three plots show the time averaged SIT for the three sea-ice products. The red contours represent the sea-ice edge.**

The spatial patterns of SIT in September (Fig.9) show anomalously thick ice (>3m) in some CMIP6 models primarily in *two* regions: an elongated *tongue of thickest sea-ice* extending northward from the northwest Weddell Sea along the Antarctic Peninsula and, the other is *around the sea-ice edge*. The distinctive tongue-like pattern may be attributed to a prominent feature in the Weddell Sea called the *Weddell Gyre* (Vernet et al., 2019). This mechanism contributes significantly to the regional sea-ice dynamics in the form of an apparent westward ice motion in the southern Weddell Sea. As a result, ice convergence occurs in the southwestern Weddell, causing dynamic thickening (Shi et al., 2021). Multiple models, including IPSL-CM6A-, EC-Earth3-, NorCPM1, and ACCESS-models, which captured this tongue of thick sea-ice, also effectively captured the Weddell Gyre through their well-developed sea-ice velocity vectors (not shown). This is one of the primary reasons why these models simulate excessively thick ice along the Peninsula (Li et al., 2021). A similar spatial pattern in SIT was also observed using ICESat measurements (Holland & Kwok, 2012) and in simulations from an ocean-sea-ice model in the Southern Ocean (Holland et al., 2014).

The other region of thick sea-ice bias is the *sea-ice edge* (Fig.9). It's interesting to note that the CMIP6 models that have exhibited better performance in simulating Antarctic SIA, such as CESM2-, NorESM2-, and ACCESS-models (Holmes et al., 2019; Li et al., 2023; Roach et al., 2020; Uotila et al., 2014) show positive thickness anomalies/biases all around the sea-ice edge during September, north the 15% SIC interval. A potential explanation for this could be through combinations of changes in air-ice drag and the direction of cold or warm-air advection. These may result in northward wind stress causing the sea-ice to drift, transport and accumulate causing

dynamic convergence at the sea-ice edge (Singh et al., 2021; Holland et al., 2014; Holland & Kwok, 2012). Another reason could be the high-intensity ocean-wave fields linked to the Southern Ocean which deeply infiltrate the marginal ice zone. This penetration induces alterations in thickness distribution through processes like rafting and ridging, especially in the vicinity of the ice edge (Langhorne et al., 1998). In any case, the simulated sea-ice at the ice edge is much thicker than observed and further study is required to eliminate modeling error as its cause.

Among the CMIP6 models, there are two clear patterns that emerge across the smaller subsets (Fig.9): *First*, a general "thin" bias in the models which is observed throughout the ice pack at all longitudes, coupled with a "thick" bias in the marginal ice zone/ice edge (as seen in CESM2, NorESM2, and ACCESS), likely reflecting the net northward transport of Antarctic sea-ice. *Second*, another smaller subset of models (such as IPSL, SAM0-UNICON, NorCPM1, CESM2, and CanESM5) which exhibit more zonally assymmetric biases, providing distinct signatures of the Ross and Weddell gyres—particularly in the form of thick ice at the outflow and thin ice at the inflow. Both modeled patterns underscore the importance of sea-ice dynamics and invites further investigation.

Additionally, Fig.9 and S1-S3 depict varying model spreads in SIT simulations with noticeable differences in their spatial biases and distributions. These variations are largely driven by differences in parameterization schemes and the representation of underlying physical processes across the models. While these discrepancies affect the accuracy of SIT simulations, a detailed examination of each model's parameterization and physics is beyond the scope of this study. Our primary focus is on the broader evaluation of the spatial distribution of SIT across the models and their comparison to observational and synthesis datasets.

In summary, all models tend to underestimate SIT and produce relatively thinner sea-ice during both September and February. These negative biases are more pronounced in September and much less pronounced in February. When comparing among the reference products, models simulate SIT closer to the reanalysis reference, i.e. GIOMAS, in both months with about 50% of the CMIP6 models (20 out of 39) having their mean spatial biases between +/-0.5m. Notably, one of our comparisons uses satellite products, which exhibit some uncertainties in estimating SIT in the Southern Ocean, particularly showing thicker values in September. Therefore, models displaying even greater positive biases (>1m) in September compared to the satellites (Fig.S1) may be simulating excessively thick sea-ice, potentially presenting a false picture of future Antarctic sea-ice changes.

## 4. Discussion

Antarctic sea-ice thickness (SIT) exhibits a seasonal cycle different from other surface sea-ice parameters with its maxima observed in February, when only the thickest multi-year ice survives the melt season. CMIP6 models generally capture this behavior, successfully simulating peak thickness in February. This agreement at summer maxima suggests that model thermodynamics allow a remnant of thick ice to persist through austral summer, similar to observations. During winter growth, however, Antarctic ice remains relatively thin on average. The thickness minimum occurs in late fall or early winter, depending on the type of SIT dataset.

Besides the thermodynamic factors, the observed discrepancies in the CMIP6 models with different sea-ice products can be due to a variety of factors such as model parameterizations and

tuning, snow ice interactions or other sea-ice processes. CMIP6 models generally capture the spatial pattern of thicker coastal ice versus thinner ice elsewhere, but they tend to underestimate the absolute thickness in these deformed-ice regions (~58% less than satellite-based estimates; Hou et al., 2024). This bias points to limitations in model dynamics: if ridging schemes or ice strength parameters are not adequately representing pressure ridge formation, models will produce too little thick ice. Moreover, the ice thickness distribution in models is sensitive to mechanical parameters– e.g. the compressive strength and the number of thickness categories –which control how readily ice converges into thick ridges (Nie et al., 2023). Numerical experiments confirm that increasing the number of ice thickness categories or a higher horizontal model resolution improves the simulation of thick ice: more categories allow a better representation of high-thickness tails from ridging and finer grids reduce artificial numerical diffusion of ice thickness.

Model parameter tuning can also impact thickness. Sea-ice models are often tuned to get realistic ice extent or concentration, which can indirectly affect SIT. For instance, adjusting albedo or ice strength to match observed ice area might cause models to grow ice either too thick or too thin as a side effect. Similarly, the GIOMAS reanalysis (with ~0.8° grid and concentration assimilation) is also known to underestimate thickness in deformed ice areas, partly due to smoothing and assimilation removing thick ice (Liao et al., 2022). This underestimation is echoed in many CMIP6 models. Studies of Arctic sea-ice in models have shown that tuning ice strength and improving ice drift physics can reduce biases in ice thickness and extent. In the Antarctic, this means capturing features like the persistent Weddell Gyre (which retains most of the multi-year ice) and properly accounting for wave-induced ice break-up and floe size effects in the outer pack. Improving sea-ice dynamics in models (ridging parameterizations, drag coefficients, and interactions with ocean currents) is essential for simulating the observed thick ice in convergence zones. Model biases in atmosphere or ocean state can also translate into SIT biases, depending on how the sea-ice model is tuned. Therefore, diagnosing thickness biases requires careful attention to model configuration: ice strength parameters, snow thermal conductivity, melt pond and albedo schemes (both in the Arctic and Antarctic), and ocean mixing coefficients all may be tuned to some degree. Going forward, studies recommend reducing the need for ad-hoc tuning by improving physical realism– for example, ensuring accurate coupling of ice with atmosphere/ocean momentum and heat fluxes (Nie et al., 2023). In Arctic simulations, better coupling (e.g. improved ice–ocean drag and air–ice drag) was shown to reduce sea-ice bias, which may apply in the Antarctic as well.

High snowfall rates on thin sea-ice can lead to snow loading and flooding: the weight of snow depresses the ice floe, allowing seawater to inundate the snow layer, which subsequently freezes into snow-ice. Observations indicate that the snow-ice formation occurs across all Antarctic sectors during the growth season, contributing significantly to total ice thickness (Maksym and Markus, 2008). Climate models that include this process can build additional ice mass on top of the original ice, thickening the ice cover despite the insulating effect of snow. If a model omits or poorly parameterizes snow-ice formation, it will likely underestimate SIT, especially in heavy snowfall regions. At the same time, a deep snowpack insulates the ice from cold air, reducing basal ice growth. The net effect on SIT is a balance: more snow can mean slower growth, but also more snow-ice added. Model sensitivity studies confirm that uncertainties in snow processes strongly affect simulated SIT. For instance, altering the prescribed snow depth or thermal properties can change modeled SIT by about 10cm to nearly a meter (Nie et al., 2023). As mentioned in Sect. 2.1, snow cover is also a key factor in satellite thickness retrievals, which helps explain why models compare differently to satellite observations versus reanalyses in opposite seasons.

The presence of model deviations can hamper our understanding of climate-sea-ice interactions as
well as biological feedback between the oceans and climate. For instance, due to the existing
relationships between SIT and sea-ice motion, biases in the simulated thickness will also affect the
dynamics in the models which in turn will impact our understanding of the overall Antarctic sea-
ice trends in the models (Lecomte et al., 2016; Sun and Eisenman, 2021). Additionally, lower SIT
could create the misleading impression of lower albedo and increased light penetration,
subsequently leading to increased Primary Production (Jeffery et al., 2020). Our study does not
explore the reasons behind such continued biases in CMIP6. However, there are a variety of
potential explanations (besides ones explained above) which may include cloud effects (Kay et al.,
2016; Zelinka et al., 2020), spatial resolution that does not permit eddies, which are understood to
be highly important for representation of Southern Ocean dynamics (Poulsen et al., 2018; Rackow
et al., 2019), models lacking grounded icebergs as landfast ice (Fraser et al., 2023), biases in
Southern Ocean stratification (Martinson and Iannuzzi, 1998), and temperature (Luo et al., 2023)
and, the lack of coupled ice sheet interactions, which have relevance for the entire Antarctic
climate system (Bronselaer et al., 2018; Golledge et al., 2019; Purich & England, 2023).

## 5. Conclusions

Given the current context of recent extreme sea-ice loss, it is imperative to develop predictions
regarding Antarctic sea-ice behavior to enhance our understanding of its future variability and
response to climate change. For this we need reliable SIT estimates along with the surface sea-ice
variables, to assess the absolute  changes in the global sea-ice cover. However, due to the lack of
long-term, high-quality observation datasets, assessing Antarctic SIT and its climatic response
remains challenging.

While GCMs offer a valuable solution to the above challenge, there is an understanding that they
do not yet accurately simulate SIT and SIV. However, it is still necessary to see how well they
perform even if only to understand where more work is needed. Therefore, despite existing
limitations, this study undertook a comprehensive evaluation of Antarctic SIT and SIV by
comparing *historical* simulations of 39 CMIP6 model outputs with three different sea-ice products.
This comparison demonstrates that models can provide longer timescales of SIT data, which, when
compared with observations-based sea-ice estimates (and accounting for their limitations), can
enhance our understanding of Antarctic sea-ice. To place the historical findings in a broader
climate context, we examined future projections under the high-emission *SSP5-8.5* scenario
(2015–2100). The projections indicate pronounced SIT declines confined primarily to the cooler
seasons (winter and spring), whereas SIV and SIA exhibit a persistent, year-round decrease, with
negative anomalies intensifying after approximately 2060. These results provided valuable
insights into how the observed post-2016 declines may evolve in a warming climate and help link
present-day variability to long-term Antarctic sea-ice vulnerability. Notably, the inclusion of
simulated SIT and its variability in our analysis also reveals complex and, at times, unexpected
behavior that does not always mirror changes in SIA or volume, underscoring the need for a more
nuanced understanding of Antarctic sea-ice responses in a warming climate.

An accurate modeling of climatological mean sea-ice cover in the GCMs is an initial step and a
necessary condition for accurate projections (Holmes et al., 2022). In line with this, our study
shows that most CMIP6 models can simulate the timing of annual cycles of all the sea-ice variables
much like the sea-ice products. For SIT, the greatest agreement is observed during the maximum

in February, when the Southern Ocean retains the thickest sea-ice, consisting of very thick ice that survives the summer. However, models fail to capture the SIT minima as observed in the satellite products and instead align more closely with the synthesis estimates (GIOMAS and GECCO3), with the minima of both models and synthesis estimates occurring in May. For SIV, the CMIP6 MMM-based annual maxima are lagged by 1 and 2 months compared to the satellite products and GIOMAS, respectively. The closest agreement in the annual cycles between models and sea-ice products is seen in SIA. Despite this alignment, when examined numerically, the models substantially underestimate the annual cycles across all the variables, with relatively lower biases occurring in SIA. Biases in the modeled cycles of SIV and SIA are higher in April-October, with greater inter-model spreads in fall-winter. Conversely, SIT inter-model spreads are higher during November-March but exhibit relatively lower biases compared to the satellite dataset. CMIP6 models continue to simulate negative trends in Antarctic SIT/SIV, contrary to the observed positive trends, until mid-2015. Additionally, we observe positive trends in SIT/SIV during the cooler seasons, which are absent in SIA. These positive trends may be due to intensified seasonal winds during the cooler seasons and further imply that the variability in total sea-ice mass in these months could be influenced by thickness/volume changes. We also examined seasonal variations in sea-ice correlations, showing positive (negative) relationships between SIA and SIT during September (February). Such seasonal covariances suggest that surface sea-ice parameters such as SIA may be weak predictors of SIT. This in turn can have significant implications for our understanding of absolute sea-ice changes in the Antarctic when based solely on SIA. Investigating the reasons for such covariances is outside the scope of this study.

This study compares the SIT biases using the multi-model means, where we show that model results are closer to the reanalysis/synthesis datasets both over both space and time. The spatial variability of sea-ice biases in CMIP6 models highlights certain challenges in accurately capturing both SIT and SIC, especially near the ice edges. Models exhibit greater disagreement in regions influenced by dynamic processes like ice advection, melting, and freezing, while better agreement is seen in areas with multi-year sea-ice and inner ice packs. The variability in SIT biases is found to be closely linked to uncertainties in the model representation of the ice edge, suggesting that improvements in capturing both the dynamic processes and the correct ice edge location could enhance model performance in simulating sea-ice cover in these areas.

While many CMIP6 models simulate spatial SIT patterns similar to the three sea-ice products, they generally tend to underestimate SIT, especially during September. Intriguingly, certain models display anomalously thick sea-ice along the Peninsula and around the sea-ice edges, even greater than the reference dataset. A potential explanation for the observed thick ice in the Weddell can be the presence of fast-ice (i.e., sea-ice pinned to the coast or grounded icebergs). Such thicknesses observed around the Antarctic Peninsula in Fig.9 significantly exceed what is expected from atmospheric heat loss alone, suggesting the presence of fast ice (Fraser et al., 2023). However, we note that the GCMs do not simulate such landfast ice prognostically. Hence, accumulation of thick ice in this region, as depicted in the models, is likely driven by dynamic processes such as winds or drift, leading to ice piling up against the Antarctic Peninsula. Considering the above findings, we anticipate that future studies will investigate these aspects with respect to Antarctic SIT. Addressing such model biases could be initial steps in further improving the representation of dynamic processes in sea-ice, climate, and biogeochemical models, ensuring their accurate predictions. Understanding the biases in sea-ice parameters and physical mechanisms behind these constraints will lead to improvement in the reliability of sea-ice projections and increase confidence in our understanding of what controls the rate of Antarctic sea-ice loss. Therefore, our research

addresses a critical knowledge gap of understanding and modeling of Antarctic SIT and the
dynamics involved in shaping its temporal and spatial distributions using the long-term coupled
climate simulations.

## Author contributions

ST and MR developed the concept of the paper. ST analyzed all the data and wrote the first draft
of the paper. WH helped with the methodology and analysis of the CMIP6 modeled data. All
authors assisted during the writing process and critically discussed the contents.

## Competing interests

The authors declare that they have no conflict of interest.

## Data Availability Statement

The satellite products used in the study are available at
https://catalogue.ceda.ac.uk/uuid/b1f1ac03077b4aa784c5a413a2210bf5 for Envisat and at
https://catalogue.ceda.ac.uk/uuid/48fc3d1e8ada405c8486ada522dae9e8 for CryoSat-2 (Hendricks
et al., 2018a, 2018b). The GECCO3 sea-ice thickness data are available at https://www.cen.uni-
hamburg.de/icdc/data/ocean/easy-init-ocean/gecco3.html last access: 31 May 2021, Köhl, 2020).
The GIOMAS sea-ice thickness data are available at
https://psc.apl.washington.edu/zhang/Global_seaice/data.html (last access:26 December 2020,
Zhang and Rothrick, 2003). Monthly values of sea-ice concentration from NSIDC are available at
https://nsidc.org/data/nsidc-0079/versions/3. All the CMIP6 model datasets are available at ESGF
website: https://aims2.llnl.gov/search/cmip6/ (Table S2).

## Acknowledgement

743 M.N. Raphael and S. Trivedi acknowledge funding by the National Science Foundation (NSF)
under the Office of Polar Programs (NSF-OPP-1745089). W.R. Hobbs acknowledges support by
the Australian Government as part of the Antarctic Science Collaboration Initiative program and
receives funding from the Australian Research Council Discovery Project (DP230102994).

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
