# Peer review of "An Assessment of Antarctic Sea-ice Thickness in CMIP6 Simulations with Comparison to the Satellite-based Observations and Reanalyses"

_EGUsphere, 2024_

## Referee Comment (RC1)

**Comment on "An Assessment of Antarctic Sea-ice Thickness in CMIP6 Simulations with Comparison to the Satellite-based Observations and Reanalyses" by Shreya Trivedi et al.**

This study evaluates Antarctic sea-ice thickness (SIT) and volume (SIV) from 39 CMIP6 models from 2002 to 2014, utilizing three sea-ice products, including satellite observations and two reanalysis datasets. It analyzes spatio-temporal variations in SIT and examines the seasonal co-variability between sea-ice area (SIA) and SIT, noting a negative relationship in summer and a positive winter one. Overall, the paper is well structured and relevant, though minor revisions are recommended for clarity and insight before publication in *The Cryosphere.*

**General Comments:**

1. The manuscript outlines the sea ice products in the data and methods section. However, the details regarding each product's parameters used in this study are inadequately described, and the process of calculating the SIV and SIA simultaneously. It is essential to provide a clearer explanation of the conversion procedure and the parameters involved in obtaining the target variables. Ensure that all key variables (SIT, SIV, and SIA) are clearly defined in the calculation process in the manuscript.

2. The analysis shows that CMIP6 models better match satellite observations at thickness maxima (February) and reanalysis at minima. I recommend that the authors discuss potential causes, for example, whether the models' parameterizations, snow-ice interactions, or ice growth processes could be driving the observed discrepancies.

3. The description and analysis of Figure 2 are confusing. The manuscript mentioned that "we observe positive trends in SIT/SIV during the cooler seasons, which are absent in SIA." However, the trend in the description of Figure 2 is not clear regarding which data's trend is being referenced. Meanwhile, it seems that there are also positive trends in SIA during the cooler seasons, as indicated in the last figure in the fourth line.

**Minor Comments:**

Some figures do not have specific sub-diagram serial numbers, which makes it impossible to properly match the description with the diagram when describing the diagram (e.g., Figure 2). It is recommended to add an ordinal number to each subgraph as well as additional annotations such as legend. Meanwhile, the quality of some figures needs to be optimized. For example, the black squares in Figure 3 are not clearly shown.

---

## Author Comment (AC2)

**RESPONSE TO THE REVIEWERS**

We express our sincere gratitude to the editor and both the reviewers for dedicating their time & thoughtful consideration, along with providing valuable feedback and suggestions for improvement of our manuscript.

Throughout this document, reviewer comments are indicated by plain blue text and our responses are presented in plain black text, addressing each comment individually.

**REVIEWER #1**

This study evaluates Antarctic sea-ice thickness (SIT) and volume (SIV) from 39 CMIP6 models from 2002 to 2014, utilizing three sea-ice products, including satellite observations and two reanalysis datasets. It analyzes spatio-temporal variations in SIT and examines the seasonal co-variability between sea-ice area (SIA) and SIT, noting a negative relationship in summer and a positive winter one. Overall, the paper is well structured and relevant, though minor revisions are recommended for clarity and insight before publication in The Cryosphere.

**General Comments:**

1. The manuscript outlines the sea ice products in the data and methods section. However, the details regarding each product's parameters used in this study are inadequately described, and the process of calculating the SIV and SIA simultaneously. It is essential to provide a clearer explanation of the conversion procedure, and the parameters involved in obtaining the target variables. Ensure that all key variables (SIT, SIV, and SIA) are clearly defined in the calculation process in the manuscript.

Thank you for your comment.

We have updated our manuscript, and it now clearly outlines the details regarding each of the used sea-ice product. It provides clearer explanations of the conversion procedure and the parameters involved in obtaining the target variables such as SIA and SIV. The following lines and text have been added under each dataset section to address the above comment:

**Lines 113-118:**

"To eliminate mismatch in the spatial resolution, all the sea-ice products were regridded onto a common CMIP6 model grid. SIA is calculated by multiplying the monthly values of sea-ice concentration (SIC) by the corresponding grid cell area and summing over the Southern Hemisphere. SIV was computed as the product of the actual floe thickness, SIC and the grid cell area, summed over the circum-Antarctic region."

**Lines 140-144:**

For the satellite product, SIA is calculated by multiplying the SIC with the corresponding grid cell area, both derived from the reanalysis dataset provided by the National Snow and Ice Data Center (NSIDC; Comiso, 2017) for the period 2002-2014. SIV is then computed by multiplying the resulting SIA with the satellite-derived floe thickness and summing over the entire Southern Hemisphere.

**Lines 176-179:**

For the reanalysis/synthesis datasets, SIA is calculated as the product of the SIC and the corresponding grid cell area obtained from the respective datasets. The SIA value so obtained is then multiplied with the "effective thickness" from each of the reanalysis/synthesis dataset to obtain their respective SIV.

2. The analysis shows that CMIP6 models better match satellite observations at thickness maxima (February) and reanalysis at minima. I recommend that the authors discuss potential causes, for example, whether the models' parameterizations, snow-ice interactions, or ice growth processes could be driving the observed discrepancies.

Thank you for your comment. Our updated manuscript now includes a new section (Sec. 4), Discussions, which discusses the details and potential causes for why models simulate SIT differently across different seasons/months.

The following section has been added to address the above comment. **Lines 536-604:**

[revised manuscript text omitted]

Thank you for pointing out the confusion in the text. We have updated the lines in the section to avoid any further confusion with our statements. See the text below. Furthermore, we have also updated the figure, caption and serial numbers of Figure 2 to improve its clarity.

**Lines 278-285:**
Next, we look at the seasonal trends in the anomalies of the three sea-ice variables (SIT, SIV and SIA) across different CMIP6 models and the sea-ice products. In general, CMIP6 models simulate negative trends in Antarctic SIA and extent (Roach et al., 2020 and Shu et al., 2020), which contrast with the observed positive trend until mid-2015 (Li et al., 2023; Shu et al., 2015; Turner et al., 2013). Figure 2 demonstrates a similar pattern in the simulated SIT/SIV with a negative trend noticed across all the sea-ice variables. However, differences emerge in seasonal trends for

different sea-ice products (blue lines for GECCO3; green for GIOMAS and red for the satellite products in Fig.2) starting in the early 2000s.

In particular, significant positive trends in SIT/SIV are observed during the cooler seasons (winter and spring; Fig.2g-h,j-k), while no such trends (or very weak trends in SIV) are detected in the warmer seasons (Fig.2a-b,d-e). For SIA, small positive trends are observed during the warmer seasons (Summer and Fall), consistent with the previous studies (Martinson, 2012; Eayrs et al., 2019), while these trends are markedly reduced during the cooler seasons. This reduction is attributed to the thermodynamic constraints imposed by the Southern Ocean polar front, which limits ice edge expansion during the colder months. The observed positive trends in SIT/SIV during winter and spring imply that changes in Antarctic SIT may contribute to variability in total sea-ice mass/volume during these colder months. This could be linked to the presence of robust land-ocean temperature gradients during winters, which may result in high-intensity winds, one of the significant contributors to SIT/SIV fluctuations in the Southern Ocean (Zhang, 2014).

**Minor Comments:**

Some figures do not have specific sub-diagram serial numbers, which makes it impossible to properly match the description with the diagram when describing the diagram (e.g., Figure 2). It is recommended to add an ordinal number to each subgraph as well as additional annotations such as legend. Meanwhile, the quality of some figures needs to be optimized. For example, the black squares in Figure 3 are not clearly shown.

Thank you for pointing out this detail. Figures 2 and 3 and their captions are updated based on the comments provided.

[Figure]

**Figure 3: Taylor Diagrams representing spatial correlation and standard deviation using time-averaged means between CMIP6 models and different sea-ice products. For each model and synthesis product, two statistics are plotted: the Pearson correlation coefficient, related to the azimuthal angle (blue contours); and the standard deviation viz proportional to the radial distance from the origin (black dotted contours). Black solid contour corresponds to the standard deviation of the reference dataset. Red dots represent individual CMIP6 models and green stars represent synthesis datasets. The period used for comparison is 2002-2014 for February (a,b,c) and September (d,e,f). For February (a and b), GECCO3 is not included as it had very small negative correlation coefficients. Reference datasets used for SIT/SIV and SIA are Envisat-CryoSat-2 and NSIDC, respectively. Higher correlation coupled with a lower RMSD indicates greater accuracy of CMIP6 models in simulating the sea-ice variables.**

This study provides a systematic evaluation of Antarctic sea ice simulations in CMIP6 models by integrating satellite observations and reanalysis data. The paper is well-structured, employs rigorous methodology, and presents comprehensive data analysis. It contributes meaningfully to the field of Antarctic sea ice modeling.

**Below are my detailed review comments:**
1. The manuscript mentions GIOMAS and GECCO3 reanalysis products but does not explicitly clarify whether these products assimilate observational data (e.g., sea ice concentration, satellite retrievals).

Thank you for pointing it out. We have added product details for both the reanalysis/synthesis products (GIOMAS and GECCO3). The updated lines now provide information about the data assimilations used in the above sea-ice products.

**Lines 155-173:**

GECCO3 ocean synthesis is an improved version of GECCO2 based on MITgcm which employs the adjoint method to fit the model to a large variety of data over a multidecadal period (1948-2018). Unlike GECCO2, it is optimized over only one assimilation window. GECCO3 has 40 levels and uses the horizontal and vertical grid of the ocean component of MPI-ESM in the MR/HR configuration, providing a global eddy-permitting synthesis at a nominal resolution of 0.4° (Köhl, 2020). While GECCO3 does not directly assimilate sea-ice data, its ocean-ice coupling is influenced by the model's oceanic state. Hence, this synthesis primarily assimilates oceanic data, including temperature and salinity (EN4.2.1), along-track sea-level anomalies (AVISO), and sea surface temperature (HadISST), using a 4D-VAR (adjoint) method (Köhl, 2020).

GIOMAS uses the Parallel Ocean Model coupled with a 12-category thickness and enthalpy distribution ice model at a horizontal resolution of 0.8° (Zhang & Rothrock, 2003). In GIOMAS, the modeled SIC is nudged by assimilating satellite derived SIC from the Special Sensor Microwave Imager (SSM/I) launched by the Defense Meteorological Satellite Program (Weaver et al., 1987). Subsequently, other modeled variables including SIT are adjusted accordingly. This

process reduces the root-mean-square difference and improves the correlation between modeled SIT and observed SIT, while also causing the thinning of the mean SIT. This assimilation method has demonstrated good agreement of modeled SIT with satellite observations in the Arctic (Lindsay & Zhang 2006) and is useful for studying long-term variations in Antarctic sea-ice (Liao et al., 2022; Shi et al., 2021).

2. Lines 211–213 mention that the IPSL and EC-Earth3 models produce anomalously thick ice, but there is no explanation of why these models behave differently (e.g., ocean/ice dynamics, resolution, parameterizations). I suggest adding a paragraph comparing key models, such as IPSL vs. CESM2, to highlight how differences in physical schemes (e.g., ridging, thermodynamics) lead to divergent sea ice thickness simulations.

Thank you for the comment. As suggested, our updated text includes a paragraph which outlines potential reasons for the different SIT simulations in IPSL and EC-Earth3 and CNRM models.

**Lines 231-243:**

A few models, namely the IPSL and EC-Earth3-models (characterized by significant warm Southern Ocean biases; Döscher et al., 2022), exhibit anomalously thick sea-ice (>3m) in February. This can be attributed to the dynamically-related reasons causing the drift and deformation of the ice. This thick ice is captured in these models mainly due to the well-resolved wind patterns (Vancoppenolle et al., 2009) which result in the sea-ice drift and accumulation of the multi-year ice which survives the summer melt and is found mainly in the Weddell Sector during February. Additionally, in models with NEMO-LIM3 sea-ice modules, such as IPSL- and the EC-Earth3-models, the frazil ice formation dominates the total ice growth which may again lead to an accumulated SIT during February (Lie et al., 2021). By contrast, CNRM-models exhibit anomalously low thickness throughout the year. This is primarily due to high negative biases in their SIC simulations and their inability to realistically simulate thick sea-ice during the austral summer in the Weddell Sea (Voldoire et al., 2019).

Additionally, we have now added a new section (Sec. 4: Discussions) which outlines the detailed description of the potential reasons for seasonal differences in SIT simulations in different CMIP6 models.

3. Although the paper discusses biases in CMIP6 models when simulating Antarctic sea ice, the analysis of the underlying causes of these biases is somewhat brief. I recommend a more detailed discussion of the sources of model discrepancies, particularly with respect to how different models perform in different regions and the specific impacts these factors have on sea ice simulations.

Thank you very much for the inquiry. Our updated manuscript now includes a new section (Sec. 4), Discussions, which discusses the details and potential causes for why models simulate SIT differently across different seasons/months.

Following section has been added to address the above comment. **Lines 536-604:**

**4. Discussion**

[revised manuscript text omitted]

5. The right column of Figure 2 (SIA anomalies) excludes GECCO3 and GIOMAS, unlike the left and middle columns (SIT/SIV). I suggest either adding the SIA data from

Thank you for this keen observation and your comment. We have updated our figure which now includes SIA data from GECCO3 and GIOMAS.

[Figure]

[Figure]

**Figure 2: Anomalies for four seasons: Summer and Fall (Warm Seasons; a-f), Winter and Spring (Cold Seasons; g-l) of SIT (left), SIV (middle) and SIA (right) of the circumpolar Antarctic. All the CMIP6 models are shown as grey lines, Multi-model mean in dashed line, GECCO3 in blue, GIOMASS in green, and NSIDC/ Envisat-CS-2 in red. All the sea-ice products extend for the time-period 1979-2014 except for satellite products which are only available for 2002-2014.**

6. The analysis stops at 2014, while Antarctic sea ice has shown a significant decline since 2015 (Raphael & Handcock, 2022). It would be helpful to briefly discuss whether CMIP6 models (e.g., SSP scenarios) capture this reversal, even if reanalysis data does not include post-2014 information.

Thank you for your keen interest in the analysis during the future scenarios. We have included a supplementary figure for the same (Fig. S1) and have discussed it in our manuscript as well.

[Figure]

[Figure]

**Figure S1: Anomalies for four seasons: Summer and Fall (Warm Seasons), Winter and Spring (Cold Seasons) of SIT (left), SIV (middle) and SIA (right) of the circumpolar Antarctic. All the CMIP6 models are shown as colored lines, and the Multi-model Mean in black line. The time-period is 2015-2100 for the SSP585 scenario. Since sea-ice products end in 2014, they could not be included here. Note that 3 models (namely ACCESS-CM-2, ACCESS-EM2-1 and CESM2-WACCM show slightly higher anomalies (especially for SIA), particularly evident during the cooler seasons (lower two panels).**

**Lines 297-303:**

We further examined seasonal anomalies in sea-ice variables under the high-emission SSP5-8.5 scenario for the period 2015–2100 (Fig. S1). The results are consistent with those in Fig. 2, showing pronounced seasonal trends in SIT, particularly during the cooler seasons (winter and spring). Under the warmer scenario, all CMIP6 models project a significant decline in SIT during these cooler months, while trends in SIT are largely absent during the warmer seasons (summer and fall). In contrast, SIV and SIA exhibit a persistent year-round decline, with negative anomalies becoming more pronounced from approximately 2060 onward (Fig. S1).

7. The Taylor diagrams in Figure 3 use "satellite products" as the reference, but it is unclear whether Envisat-CryoSat-2 is averaged or used separately. I recommend explicitly stating which dataset is used as the reference for each variable and justifying any averaging.

Thank you for this clarification comment. We now explicitly state in our text how the reference datasets are used for Taylor Diagram calculations. Additionally, we have also updated Fig.3 and its caption to reflect the same.

**Lines 312-315:**

We use the satellite dataset as the observation reference for SIT and SIV, calculating RMSDs and correlation values across the grids. For SIA, the reference dataset used is NSIDC. All the calculations were performed based on area-integrated spatial averages of reference datasets of SIT, SIV and SIA over the circum-Antarctic for February and September.

8. Section 3.4 relies on GIOMAS as the primary reference due to satellite biases in September. However, GIOMAS shows large errors in February (Figure 1c, early maxima). I suggest acknowledging the limitations of GIOMAS and cross-validating key results with GECCO3 and satellite data where possible.

Thank you for the comment. We have updated the first paragraph of Sec. 3.4 to accommodate the above comment. The following lines have been added for the same:

**Lines 446-466:**

In this section, we perform a spatial comparison for circum-Antarctic SIT across 39 individual CMIP6 model ensembles using different sea-ice products as observational references. However, we only discuss in detail GIOMAS as the observational reference. Here, GIOMAS is selected for a detailed discussion instead of other sea-ice products, primarily due to its relative overall closer agreement with the models (See Sec.3.1) and the very high SIT biases observed in the satellite products during September (Fig.5 and S2). In general, during both the months the mean spatial distribution of SIT in models shows lower biases with respect to the reanalyses/syntheses when compared to the satellite product (See Sect.3.3). GECCO3 produced results comparable to the GIOMAS reanalysis product (not shown). Although GIOMAS displays an earlier seasonal maximum, the MMMs exhibit the lowest biases relative to GIOMAS for this month, compared to GECCO3 and satellite-derived products (Fig.5).

Spatially, the satellite as well as the reanalysis/synthesis products show that thickest sea-ice resides in the western Weddell Sea along the Antarctic Peninsula and along the coastal edges of the ABS in the form of multi-year ice (Fig.8 and S3). There is relatively thinner sea-ice observed in the eastern Antarctic (Kurtz & Markus, 2012). Our analysis reveals that most of the CMIP6 models capture a similar spatial pattern in SIT around the Antarctic however, they do exhibit biases and underestimate SIT (Fig.5 and Fig.S2-S4). In February, over half of the models simulate thinner sea ice in the Weddell Sea compared to the GIOMAS product (Fig.S3) while thin biases relative to the satellite products are simulated almost across all the models (Fig.S4). On the circum-Antarctica scale, about 38% of the models simulate thicker ice compared to GIOMAS, with only 5 out of 39 models showing SIT greater than that estimated by satellite products.

Some of the legends are not clear enough. It is recommended to directly mark them in the figures or explain them in detail in the captions.

Thank you for pointing it out. We have updated Figures 2 and 3 along with their captions to better explain the details and to avoid confusion in differentiating various metrics.

9. Although the paper proposes directions for model improvements (e.g., improving ice-edge simulation accuracy, enhancing dynamic process representations), these suggestions are rather general and lack specific implementation details.

Thank you for the comment. We acknowledge the importance of this line of inquiry.

Here, our study provides a broad evaluation and comparison of CMIP6 climate models with three distinct sea-ice products, with the primary aim of assessing overall performance rather than analyzing individual model biases or improvements in detail. As such, we only briefly address model biases and suggest general directions for improvement, without proposing specific correction strategies. However, we believe it is best that such correction strategies are pursued in future studies, dedicated to detailed model diagnostics and development.

---

## Author Response (AR2)

**Editor decision: Publish subject to minor revisions (review by editor)**

Dear Authors,

First of all, apologies that it took so long (too long) for us to get back with a decision on your manuscript. But unfortunately, the responsible editor missed his deadlines numerous times. As he never responded to our editorial inquires, we have now canceled his assignment, and I have taken over as the sea ice Chief Editor. That said, I do agree that the initial start off the manuscript and the issues with the high similarity index make proper evaluation of the manuscript a bit more difficult than most of the manuscripts we receive. I would still ask you to better clarify, in the text, potential similarities with other versions in the internet and to explain what makes the present version unique.

We are grateful about the two constructive reviews we have received, and about your replies and suggestions. Therefore, I invite you to submit a revised version accordingly.

In addition to the reviewer comments, I agree with the one reviewer that it is unfortunate that your study ends/ended in 2014, when the action only happens afterwards. I appreciate that you have provided the additional results about future developments. But I don't understand why you suggest placing it into the appendix, instead of properly integrating it in the text and addressing the changes that happened since 2017, or at least the occurrence of similar events sometime in the time series (similar to ice depletion events (?) that are discussed in the Arctic)? I suggest considering integrating these results and discussion in the main text. I am looking forward to receiving your comments and revisions.

Thank you for your patience and apologies again.

Christian Haas

**Response to Editor Comments (From the Authors):**

We sincerely thank the Chief Editor, Dr. Haas, for taking the time to handle our manuscript and for the constructive feedback. We have revised the manuscript accordingly.

1. **Regarding the similarity index:**

We acknowledge the editor's concern regarding the similarity index. The overlap primarily arises from an earlier and much less scientifically developed version of this manuscript that was uploaded to open-access preprint repositories (ESS Open Archive), following a prior submission to *AGU–Geophysical Research Letters* (submitted in October 2024). That submission was not accepted, however the preprint remained publicly available, and despite multiple requests, the ESS Open Archive has not permitted its removal. To date, none of the information in that preprint has been published after peer review. Moreover, the version previously submitted to AGU–GRL and uploaded as a preprint, primarily focused on a preliminary comparison of Antarctic sea-ice thickness (SIT) across CMIP6 models using *historical* simulations (1979–2014). **That version "provided a brief analysis of SIT trends and inter-model variability but lacked a detailed assessment of model biases, seasonal behavior, model performances in warmer scenarios and connections with other sea-ice parameters".** We include the statement in quotes above in our text, **lines 84 – 89** as recommended by the Editor, in order to help differentiate between the manuscript under consideration by *The Cryosphere* and the preprint.

The current manuscript submitted to *The Cryosphere* presents a significantly expanded and refined study. It integrates new analyses using future projection experiments (SSP5-8.5), introduces multiple new figures and sections (including seasonal bias assessments, future scenario assessments and spatial distribution analyses), and offers a much deeper discussion of the physical mechanisms driving model discrepancies (a new discussion section). Moreover, the results have been completely restructured for clarity and supported by newly added references, updated datasets, and improved visualizations. Together, these enhancements make this current paper a substantially more comprehensive and original contribution than the GRL preprint.

Following are details of the major revisions which distinguish the current manuscript from the preprint. We do not compare these revisions and new work to the preprint in the text of the manuscript under consideration since that could interrupt the flow of the arguments presented in the manuscript:

- **Expanded dataset and methodology descriptions:** We started out with historical experiments of 39 coupled climate models (CMIP6). Our revised manuscript now also includes the selected 39 models for one of the future scenarios (SSP5-8.5) for an additional analysis. The dataset section provides a detailed and updated explanation of all datasets and analytical methods used.

- **Section 3.1 expanded:** This section now includes **Figure 2** (previously in the Supplement), illustrating the seasonal trends in SIT, SIV, and SIA across CMIP6 models and sea-ice products. The accompanying analysis is substantially expanded and newly written.
- **New Section 3.2 – "Future Projections under SSP5-8.5 (2015–2100)"**: This section (with new **Figure 3**) extends the analysis beyond 2014 to evaluate projected Antarctic sea-ice changes through 2100. This addition directly responds to the editor's and reviewer's suggestions and was not present in any earlier version.
- **Revised Section 3.3 – "Seasonal Variations and Inter-relationships"**: Now includes a **Taylor Diagram (Fig. 4;** previously in the Supplement**)** with updated visualizations and a more detailed interpretation than previous drafts.
- **New Section 3.4 – "Seasonality in SIT Biases"**: This section (with new **Figures 6–8**) presents original analyses of SIT bias seasonality, which did not exist in any earlier version or supplement.
- **Revised Section 3.5 – "Spatial Distribution of SIT"**: This section now uses **GIOMAS reanalysis data** (instead of Envisat–CryoSAT-2) as the reference, with entirely new figure (**Fig. 9**), visualizations, and interpretations, making it distinct from prior versions.
- **New Section 4 – "Discussion"**: This is a completely new section, distinct from the previous "Conclusion." It provides an in-depth discussion of model parameterizations, snow–ice interactions, and ice growth processes contributing to the observed discrepancies among models, expanding on reviewer suggestions.
- **Revised Section 5 – "Conclusions"**: The conclusion section has been rewritten and renumbered. It now focuses exclusively on summarizing findings and implications, with updated phrasing and clarity.
- **Updated references:** Newly published literature relevant to our study has been incorporated throughout the manuscript.

2. **Regarding the integration of future scenario results:**

Thank you for your suggestion. We were concerned about exceeding the recommended length (and the number of figures in the main text) of the paper, hence we placed the figures for the future emissions in the supporting document and kept the discussion succinct.

However now, following your suggestion, we have moved the SSP5-8.5 future scenario results (2015–2100) from the Appendix into the main Results section (Section 3.2). We also added a brief discussion connecting these findings to post-2015 Antarctic sea-ice minima, as suggested. The following section has been added to accommodate the comments:

**Lines 304-341:**

**"3.2. Future Projections under SSP5-8.5 (2015–2100)**

Since our study uses *historical* experiments in climate models which ends in 2014 (due to the cut-off limits of the *historical* simulations), this section acknowledges the importance of assessing future sea-ice developments, particularly in the Antarctic where pronounced declines in the sea-ice have been observed since 2016 (Raphael and Handcock, 2022; Wang et al., 2022; Turner et al., 2022; Eayrs et al., 2021).

[Figure]

[Figure]

**Figure 3: Anomalies for four seasons: Summer and Fall (a-f; Warm Seasons), Winter and Spring (g-l; Cold Seasons) of SIT (left), SIV (middle) and SIA (right) of the circumpolar Antarctic. All the CMIP6 models are shown as colored lines, and the Multi-model Mean in black line. The time-period is 2015-2100 for the SSP5-8.5 scenario. Since sea ice products end in 2014, they could not be included here. Note that 3 models (namely ACCESS-CM-2, ACCESS-EM2-1 and CESM2-WACCM show slightly higher anomalies (especially for SIA), particularly evident during the cooler seasons (i and l).**

We examined the seasonal anomalies in sea-ice variables under the high-emission SSP5-8.5 scenario for the period 2015–2100 (Fig. 3). The results are consistent with those in Fig. 2, showing pronounced seasonal trends in SIT, particularly during the cooler seasons (winter and spring). Under the warmer scenario, all CMIP6 models project a notable decline in SIT during these cooler months, while trends in SIT are largely absent during the warmer seasons (summer and fall). In contrast, SIV and SIA exhibit a persistent year-round decline, with negative anomalies becoming more pronounced from approximately 2060 onward (Fig. 3).

When assessing the Antarctic sea ice distributions in the post-2014 period, it is expected that under warming scenarios, models will show reductions in sea ice owing to their response to increasing temperatures. However, our results reveal a seasonally asymmetric pattern of decline: SIA and SIV decrease persistently throughout the year, while SIT exhibits notable thinning only during cooler seasons. This indicates that the overall reduction in Antarctic sea ice projected under warmer scenarios is likely driven by sustained losses in area and volume as actual thinning is not consistently observed across all seasons. Consequently, much of the reduction in simulated sea ice likely arises from changes in surface coverage rather than from widespread structural thinning. These projected declines also correspond well with the observed post-2016 record-low Antarctic sea-ice extents (Turner et al., 2017; Schlosser et al., 2018), indicating that the recent losses may represent the early onset of the long-term downward trend simulated under high-emission scenarios. Such consistency between observations and projections highlights the increasing vulnerability of the Antarctic sea ice system to ongoing atmospheric and oceanic warming. Future studies are needed to investigate this in detail."

Following lines have also been added in the manuscript:

**Lines 199-201** in the Dataset Section:

"In addition to the *historical* experiments, we also analyze a warmer scenario, SSP 5-8.5 (Shared Socio-economic Pathway; 2015-2100) to compare with the sea-ice variability in future."

**Lines 658-668** in the Conclusion Section:

"To place the historical findings in a broader climate context, we further examined future projections under the high-emission *SSP5-8.5* scenario (2015–2100). These results provided valuable insights into how the observed post-2016 declines may evolve in a warming climate and help link present-day variability to long-term Antarctic sea-ice vulnerability."

We believe these revisions have strengthened the manuscript, and we are grateful for the opportunity to revise and resubmit.